# PRAMEL7 and CUL2 decrease NuRD stability to establish ground-state pluripotency

Meneka Rupasinghe[1,2,7], Cristiana Bersaglieri [ID][1,7], Deena M Leslie Pedrioli [ID][1], Patrick GA Pedrioli [ID][3,4], Martina Panatta[1,5], Michael O Hottiger [ID][1], Paolo Cinelli [ID][6] & Raffaella Santoro [ID][1✉]

## Abstract

Pluripotency is established in E4.5 preimplantation epiblast. Embryonic stem cells (ESCs) represent the immortalization of pluripotency, however, their gene expression signature only partially resembles that of developmental ground-state. Induced PRAMEL7 expression, a protein highly expressed in the ICM but lowly expressed in ESCs, reprograms developmentally advanced ESC+serum into ground-state pluripotency by inducing a gene expression signature close to developmental ground-state. However, how PRAMEL7 reprograms gene expression remains elusive. Here we show that PRAMEL7 associates with Cullin2 (CUL2) and this interaction is required to establish ground-state gene expression. PRAMEL7 recruits CUL2 to chromatin and targets regulators of repressive chromatin, including the NuRD complex, for proteasomal degradation. PRAMEL7 antagonizes NuRD-mediated repression of genes implicated in pluripotency by decreasing NuRD stability and promoter association in a CUL2-dependent manner. Our data link proteasome degradation pathways to ground-state gene expression, offering insights to generate in vitro models to reproduce the in vivo ground-state pluripotency.

**Keywords** PRAMEL7; Cullin 2; NuRD; UHRF1; Ground-state Pluripotency
**Subject Categories** Chromatin, Transcription & Genomics; Post-translational Modifications & Proteolysis; Stem Cells & Regenerative Medicine

## Introduction

Mouse embryonic stem cells (ESCs) are derived from the inner cell mass (ICM) and represent the immortalization of naive pluripotency. Depending on culture conditions, ESCs can acquire molecular features that are distinct from those characterizing the developmental ground-state of pre-implantation epiblast cells (Hackett and Surani, 2014). Mouse ESCs can be propagated in medium containing fetal calf serum and leukemia inhibitory factor (LIF) (ESC+serum) or in serum-free 2i medium (ESC+2i) that contains LIF plus two small-molecule kinase inhibitors for MEK/ERK (PD0325901) and GSK3 (CHIR99021) (Ying et al, 2008). Both ESC+2i and ESC+serum are pluripotent. However, ESC+serum exhibit an altered transcriptional and epigenetic profile relative to preimplantation epiblast cells and are considered to be functionally naive but not ground-state (Tang et al, 2010). In contrast, ESC+2i closely resemble the developmental ground-state in vivo (Boroviak et al, 2014). Compared to ESC+serum, ESC+2i exhibit a transcriptional profile close to E4.5 epiblast cells and a less repressed epigenetic landscape, including a hypomethylated genome that is comparable to the DNA methylation state of the inner cell mass (ICM) (Boroviak et al, 2014; Ficz et al, 2013; Habibi et al, 2013; Leitch et al, 2013; Marks et al, 2012; Smith, 2012).

We have recently shown that Preferentially Expressed Antigen in Melanoma-like 7 (PRAMEL7) is implicated in the establishment of ground-state pluripotency (Graf et al, 2017). PRAMEL7 is expressed at high levels at the morula stage and in ICM but is completely absent in post-implantation embryos and in differentiated tissues (Bortvin et al, 2003; Casanova et al, 2011; Cinelli et al, 2008). During ICM transition to ESCs, PRAMEL7 is also strongly downregulated. Further, *Pramel7* knock out (KO) embryos arrested during development at the morula stage, indicating an important role for the establishment of the blastocyst (Graf et al, 2017). Induced PRAMEL7 expression in ESC+serum caused global DNA hypomethylation and promoted a gene expression signature close to developmental ground-state (Graf et al, 2017). PRAMEL7-mediated DNA hypomethylation of ESC+serum occurs since PRAMEL7 targets for proteasomal degradation UHRF1, an essential co-factor of the de novo DNA methyltransferase 1 (DNMT1) (Bostick et al, 2007; Liu et al, 2013; Sharif et al, 2007). A similar observation was also reported during the transition of ESC+serum to ESC+2i, showing UHRF1 downregulation at protein levels (von Meyenn et al, 2016). Finally, PRAMEL7/UHRF1 expression is mutually exclusive in ICMs whereas *Pramel7*-KO embryos express high levels of UHRF1 (Graf et al, 2017). However, PRAMEL7-mediated DNA hypomethylation cannot entirely explain the significant transcriptional changes occurring upon expression of PRAMEL7 in ESC+serum. Indeed, DNA

[1]Department of Molecular Mechanisms of Disease, DMMD, University of Zurich, 8057 Zurich, Switzerland. [2]Molecular Life Science Program, Life Science Zurich Graduate School, University of Zurich, 8057 Zurich, Switzerland. [3]Department of Health Sciences and Technology, ETH Zurich, 8093 Zurich, Switzerland. [4]Swiss Institute of Bioinformatics (SIB), Lausanne, Switzerland. [5]RNA Biology Program, Life Science Zurich Graduate School, University of Zurich, Zurich, Switzerland. [6]Department of Trauma Surgery, University Hospital Zurich, University of Zurich, Rämistrasse 100, 8091 Zurich, Switzerland. [7]These authors contributed equally: Meneka Rupasinghe, Cristiana Bersaglieri.
✉E-mail: raffaella.santoro@dmmd.uzh.ch

hypomethylation observed in knockout of all *Dnmts* or *Uhrf1* in ESCs or upon culture in 2i conditions has little effect on gene expression (Ficz et al, 2013; Fouse et al, 2008; Sharif et al, 2016). These observations suggest that PRAMEL7 might have additional functions for the establishment of ground-state pluripotency that go beyond DNA methylation pathways.

In this study, we show that PRAMEL7-mediated reprogramming of ESC+serum into a developmental ground-state gene signature requires the interaction of PRAMEL7 with Cullin2 (CUL2), a core component of CUL2-RING E3 ubiquitin-protein ligase complex, which mediates ubiquitination of target proteins, leading to their degradation (Cai and Yang, 2016). We show that PRAMEL7 recruits CUL2 to chromatin and identify PRAMEL7/CUL2-targets for proteasomal degradation that are linked to pluripotency pathways in stem cells. These PRAMEL7/CUL2-targets are mainly components of repressive chromatin, such as the Nucleosome Remodeling and Deacetylase (NuRD) complex. As example of this regulation, we show that PRAMEL7/CUL2 axis contrasts the repression of NuRD-target genes implicated in pluripotency by reducing the binding of the NuRD component CHD4 at PRAMEL7-regulated genes in a CUL2-dependent manner. The results support a role of PRAMEL7 in the establishment of ground-state pluripotency, acting on the modulation of chromatin repressive factors to promote a ground-state transcriptional signature.

## Results

### PRAMEL7-CUL2 and -BC box domains are required for PRAMEL7-CUL2 interaction and UHRF1 degradation

To determine how PRAMEL7 reprograms ESCs toward a ground-state gene expression signature, we set to identify PRAMEL7 targets for proteasomal degradation that could play a role during PRAMEL7-mediated gene expression changes. Previous work showed that PRAMEL7 associates with CUL2 (Graf et al, 2017). This result prompted us to determine whether CUL2 is required to PRAMEL7-mediated gene expression in ESCs by generating PRAMEL7 mutants with impaired ability to associate with CUL2. We performed homology domain analysis with several proteins known to interact with CUL2 and containing Cul2 and BC box domains, which mediate the interaction with CUL2 and ELONGIN C respectively (Costessi et al, 2011), and identified these domains at the N-terminus of PRAMEL7 (Fig. 1A). To assess the functionality of PRAMEL7-CUL2 and -BC box domains, we introduced point mutations at conserved amino acids within the Cul2 or BC box domains or both (PRAMEL7$_{C2mut}$, PRAMEL7$_{BCmut}$, PRAMEL7$_{BC/C2mut}$) and generated a small N-terminus truncated PRAMEL7 missing the first 40 amino acids (PRAMEL7$_{\Delta N}$) that only contain the Cul2 and BC box domains, and consequentially expressing 83% of PRAMEL7 peptide (Fig. 1A,B). HA-immunoprecipitation (HA-IP) in HEK293T cells transfected with plasmids expressing HA/FLAG (H/F)-Pramel7$_{WT}$ or mutants showed that only PRAMEL7$_{WT}$ interacts with endogenous CUL2, indicating that Cul2 and BC box domains are both essential for PRAMEL7-CUL2 interaction (Fig. 1C). Western blot signals of total CUL2 (input) and PRAMEL7-interacting CUL2 (HA-IP) were also characterized by two distinct bands. CUL2 band with the higher molecular weight was previously reported to be the post-translationally modified neddylated

form that represents the active CUL2 (Duda et al, 2008; Pan et al, 2004; Wada et al, 1999). These results suggest that PRAMEL7 also interacts with active, neddylated CUL2. Next, we assessed whether the PRAMEL7-CUL2 interaction is required for the stability of UHRF1, which is known to be a PRAMEL7-target for proteasomal degradation (Graf et al, 2017) Consistent with previous results, western blot analyses showed that the expression of PRAMEL7 in HEK293T cells downregulate UHRF1 protein levels (Fig. 1D). In contrast, all PRAMEL7 mutants could not affect UHRF1 levels, suggesting a role of PRAMEL7-CUL2 interaction in regulating UHRF1 stability. To further support these data, we established ESC+serum lines stably expressing H/F-PRAMEL7$_{WT}$ or H/F-PRAMEL7$_{\Delta N}$ through the insertion of one transgene copy under the control of the CAG promoter in the *Rosa26* locus. Consistent with previous work (Graf et al, 2017), PRAMEL7 levels in parental ESC+serum are very low and under the detection limit for western analyses (Fig. 1E). In agreement with the results observed in HEK293T cells, HA-IP from ESC + H/F-PRAMEL7$_{WT}$ and ESC + H/F-PRAMEL7$_{\Delta N}$ revealed that PRAMEL7$_{WT}$ interacts with endogenous CUL2, including neddylated CUL2, while PRAMEL7$_{\Delta N}$ does not (Fig. 1E). Accordingly, UHRF1 protein levels were strongly downregulated in ESCs expressing PRAMEL7$_{WT}$ compared to parental ESCs whereas they were not affected in ESCs expressing PRAMEL7$_{\Delta N}$ (Fig. 1F). Collectively, these results indicate that PRAMEL7-Cul2 and -BC box domains mediate the interaction with CUL2 and are required for proteasomal degradation of UHRF1.

### PRAMEL7/CUL2 axis reprograms the developmentally advanced ESC+serum toward ground-state pluripotency gene expression signature

To determine whether PRAMEL7-CUL2 interaction is required to reprogram ESC+serum toward a ground-state gene expression signature, we performed RNAseq analysis and compared transcriptomic profiles of ESC + H/F-PRAMEL7$_{WT}$ and ESC + H/F-PRAMEL7$_{\Delta N}$ relative to parental ESCs (Fig. 2A,B, Dataset EV1,2). The expression of PRAMEL7$_{WT}$ in ESCs led to the upregulation of 894 genes and the downregulation of 887 genes (log$_2$ fold change ±0.58, *P* value < 0.05) (Fig. 2A, Dataset EV1). We also found significant changes in ESC+Pramel7$_{\Delta N}$ that were characterized by a higher number of downregulated genes (1046) compared to the upregulated genes (581) (Fig. 2B, Dataset EV2). By intersecting the list of genes regulated in ESC + H/F-PRAMEL7$_{WT}$ and ESC + H/F-PRAMEL7$_{\Delta N}$, we defined two classes of genes: genes that were differentially expressed only in ESC + H/F-PRAMEL7$_{WT}$ (i.e., genes depending on PRAMEL7 N-terminus) or in both ESC + H/F-PRAMEL7$_{WT}$ and ESC + H/F-PRAMEL7$_{\Delta N}$ (i.e., genes that do not depend on PRAMEL7 N-terminus) (Fig. 2C, Dataset EV3). Since PRAMEL7 N-terminus mediates the interaction with CUL2, we named the first class of genes as PRAMEL7$_{Cul2}$ (i.e., CUL2-interaction dependent genes) whereas genes regulated by PRAMEL7 independently of its N-terminus were named PRAMEL7$_{Cul2-ind}$ (i.e., CUL2-interaction independent genes). Remarkably, the large majority of genes upregulated (80%) or downregulated (60%) in ESC + H/F-PRAMEL7$_{WT}$ were PRAMEL7$_{Cul2}$ genes, indicating that the N-terminus of PRAMEL7, and most likely its association with CUL2, is the main driver for PRAMEL7-mediated gene regulation in ESCs (Fig. 2C). The other fraction of PRAMEL7-regulated genes, which represents

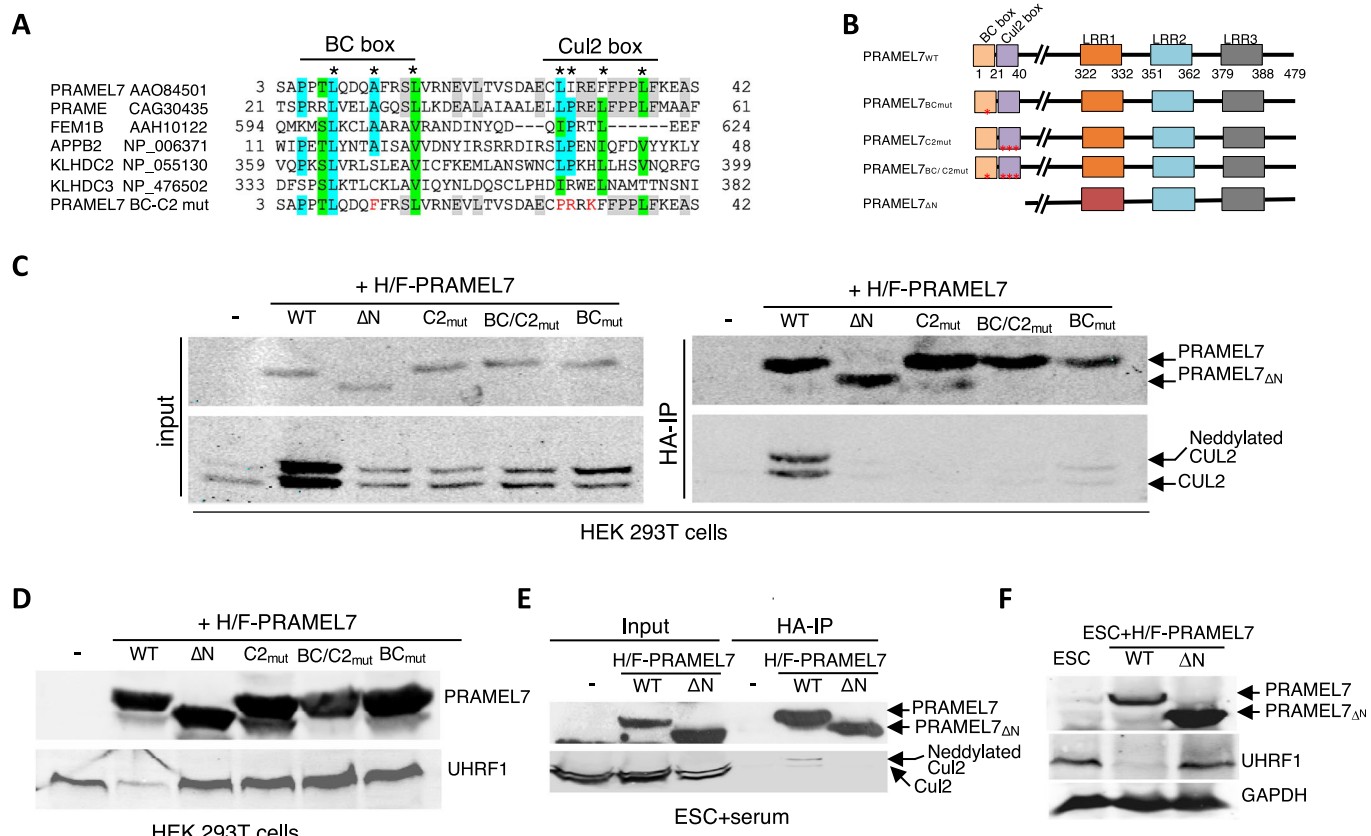

**Figure 1. PRAMEL7-Cul2 and -BC box domains are required for PRAMEL7-CUL2 interaction and Uhrf1 degradation.**

(A) Multiple sequence alignment of proteins containing Cul2 and BC box domains with PRAMEL7 showing PRAMEL7 conserved residues. Proteins aligned with identical amino acids are highlighted in cyan and those aligned with very similar ones in green. Identical amino acids between PRAMEL7 and its human homologue PRAME are in gray. Conserved PRAMEL7 residues are labeled with an asterisk. Mutated aminoacids in PRAMEL7 mutants (mut.) are labeled in red. (B) Schematic representation of PRAMEL7 domains and PRAMEL7 mutants. The N-terminal domain of PRAMEL7$_{WT}$ contain ELONGIN B and C (BC)- and Cul2 box-domains. PRAMEL7 also contains three LRR domains (Bella et al, 2008). (C) HA-immunoprecipitation (IP) in HEK293T cells transfected with HA/FLAG-Pramel7 (H/F-Pramel7) and the indicated mutant plasmids. Western blot shows PRAMEL7, CUL2 and neddylated CUL2. (D) Western blot of HEK293T cells transfected with HA/FLAG-PRAMEL7 (H/F-PRAMEL7) and the indicated mutant plasmids. PRAMEL7 and UHRF1 signals are shown. (E) HA-IP of ESC + H/F-PRAMEL7$_{WT}$ and ESC + H/F-PRAMEL7$_{\Delta N}$ lines. Western blot showing PRAMEL7, CUL2 and neddylated CUL2. (F) Western blot of ESC + H/F-PRAMEL7$_{WT}$ and H/F-PRAMEL7$_{\Delta N}$. PRAMEL7 and UHRF1 signals are shown. GAPDH serves as loading control. Source data are available online for this figure.

PRAMEL7$_{Cul2-ind}$ genes, suggests that PRAMEL7 can exert some functions independently of its N-terminus or CUL2 binding.

KEGG analysis showed that one of the top pathways enriched in upregulated PRAMEL7$_{Cul2}$ genes was signaling pathways regulating stem cell pluripotency whereas upregulated PRAMEL7$_{Cul2-ind}$ genes were enriched in pathways implicated in cancer (Fig. 2D, Dataset EV3). Downregulated PRAMEL7$_{Cul2}$ genes were enriched in pathways such as ECM-receptor interaction and RAP1 signaling that are linked to differentiation and cell migration (Li et al, 2015; Shaul and Seger, 2007; Wang et al, 2015) whereas downregulated PRAMEL7$_{Cul2-ind}$ genes showed drug and ether metabolism processes. Taken together, these results suggest that the ability of PRAMEL7 to interact with CUL2 is required to reinforce the pluripotency program and inhibit differentiation of ESC+serum. To support these data, we assessed how ESC + H/F-PRAMEL7$_{WT}$ and ESC + H/F-PRAMEL7$_{\Delta N}$ are related to embryonic stages from E3.5 to E5.5 and ESC+serum and ESC+2i by performing principal component analysis (PCA) of gene expression using our own and published gene expression profiles (Boroviak et al, 2014) (Fig. 2E).

Consistent with previous results (Graf et al, 2017), the expression of PRAMEL7 in ESC+serum reprograms the developmentally advanced ESCs toward a ground-state pluripotency signature as evident by the closeness of ESC + H/F-PRAMEL7$_{WT}$ to early embryo stages and ESC+2i. Consistent with the data shown above, this effect was much weaker in ESC+Pramel7$_{\Delta N}$, which also show high distance in the PC2 dimension, underscoring the importance of PRAMEL7-CUL2 axis to reprogram ESC+serum towards ground-state pluripotency.

## PRAMEL7 affects protein stability of regulators of repressive chromatin states through its interaction with CUL2

To determine whether PRAMEL7 could affect the stability of factors implicated in gene regulation, we performed Stable Isotope Labeling by/with Amino acids in Cell culture (SILAC) - Mass Spectrometry (MS) whole proteome analyses of parental ESCs, ESC + H/F-PRAMEL7$_{WT}$, and ESC + H/F-PRAMEL7$_{\Delta N}$

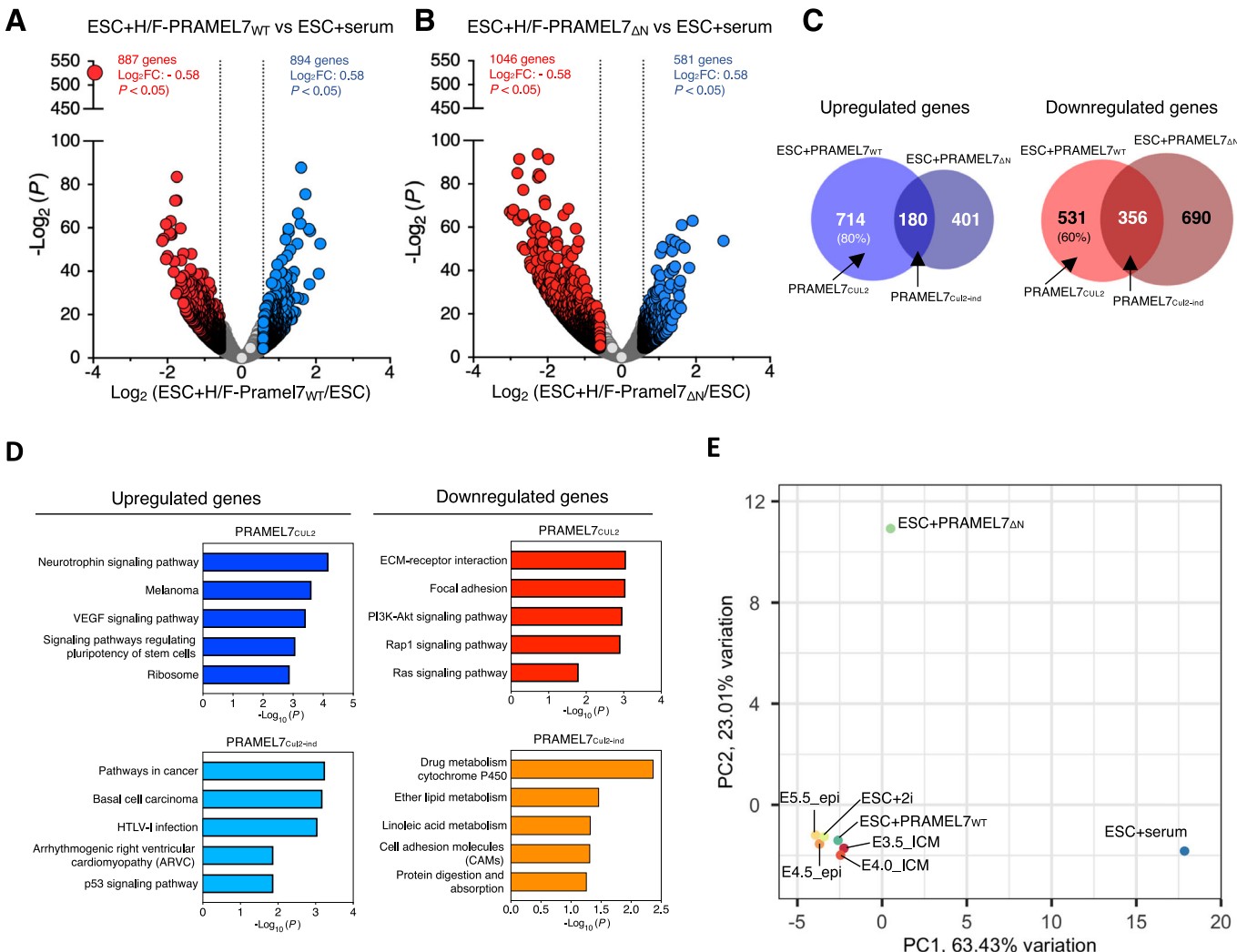

**Figure 2. PRAMEL7-CUL2 axis reprograms the developmentally advanced ESC+serum toward ground-state pluripotency signature.**

(A,B) Volcano plot showing fold change (log₂ values) in transcript level of (A) ESC + H/F-PRAMEL7_WT and (B) ESC + H/F-Pramel7_ΔN compared to ESC+serum. Gene expression values of three replicates were averaged and selected for log₂ fold changes (FC) ± 0.58 and P < 0.05. (C) Venn diagrams showing number of differentially expressed genes detected in ESC + H/F-PRAMEL7_WT vs. ESC + H/F-PRAMEL7_ΔN. Diagram highlights genes that are only regulated by the expression of PRAMEL7_WT (PRAMEL7_Cul2) and genes that are commonly regulated by PRAMEL7_WT and PRAMEL7_ΔN (PRAMEL7_Cul2-ind). (D) KEGG pathway enrichment analysis of the genes regulated by PRAMEL7_Cul2 and PRAMEL7_Cul2-ind analyzed using the DAVID annotation tool. (E) PCA showing the correlation of gene expression of ESC + H/F-PRAMEL7_WT and ESC + H/F-PRAMEL7_ΔN to ESC+serum, ESC+2i, and the early embryo stages from E3.5 to E5.5. Gene expression data from the embryos were from (Boroviak et al, 2014).

(Fig. 3A,B). To confidently identify proteins whose abundances change in response to PRAMEL7 expression in ESCs, we conducted three-forward and three-reverse SILAC experiments. In the forward SILAC experiments, parental ESCs were labeled with light isotope whereas ESC + H/F-PRAMEL7_WT and ESC + H/F-PRAMEL7_ΔN were labeled with heavy isotope. In reverse SILAC experiments the labeling was inverted. We identified 382 proteins whose abundances changed significantly (P value ≤ 0.05) in ESC + H/F-PRAMEL7_WT compared to control ESCs (183 downregulated and 199 upregulated proteins) (Fig. 3A and Dataset EV4). In contrast, only 81 proteins were significantly altered in ESC + H/F-PRAMEL7_ΔN but to a much less extent compared to ESCs expressing PRAMEL7_WT, underscoring the role of the N-terminus of PRAMEL7 to mediate the interaction with CUL2 and consequent

proteasomal degradation of target proteins (Fig. 3B). Consistent with the results described above (Fig. 1D,F), SILAC-MS measurements displayed a strong downregulation of UHRF1 protein levels in ESC + H/F-PRAMEL7_WT that were not affected in ESC + H/F-PRAMEL7_ΔN. As expected, UHRF1 downregulation in ESC + H/F-PRAMEL7_WT was only observed at protein level since RNAseq analysis did not detect significant changes of Uhrf1 mRNA levels (Dataset EV1).

Comparative analyses of the SILAC-MS and RNAseq data revealed that only 15 out of 183 proteins downregulated in ESC + H/F-PRAMEL7_WT were also significantly downregulated at transcript levels upon PRAMEL7 expression (Fig. 3A, Dataset EV4). This finding indicates that the great majority of the proteins showing reduced levels in ESC + H/F-PRAMEL7_WT were regulated at protein level.

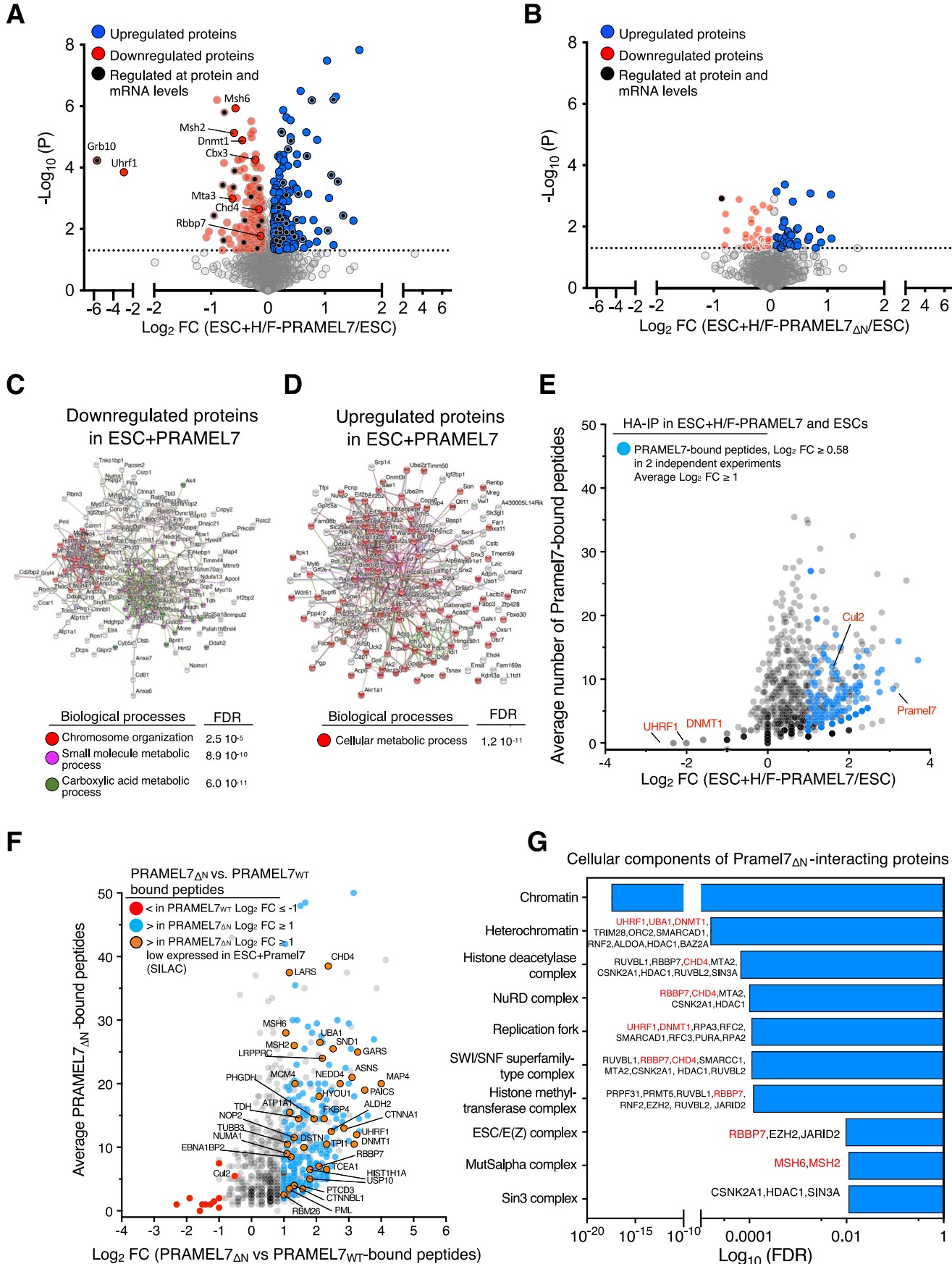

**Figure 3. PRAMEL7 regulates protein stability of chromatin regulators through its interaction with CUL2.**

(A,B) Volcano plot showing log2 fold change (FC) of protein levels measured by SILAC-MS in (A) ESC + H/F-PRAMEL7$_{WT}$ and (B) ESC + H/F-PRAMEL7$_{\Delta N}$ relative to control ESCs. Proteins regulated by PRAMEL7 were selected for a $P$ value < 0.05 and log2 fold change ±0.1 relative to ESC+serum. Measurements represent the average of three-forward and three-reverse biological experiments. Statistical significance ($P$-values) for the experiments was calculated using the paired two-tailed $t$ test. (C,D) String analyses of proteins downregulated (C) upregulated (D) and in ESC + H/F-PRAMEL7$_{WT}$ relative to control ESC that were identified by SILAC-MS. (E) Scatter plot showing log$_2$ fold changes of HA-immunoprecipitated peptides from ESC + H/F-PRAMEL7$_{WT}$ vs ESCs. Data are from two independent experiments. (F) Scatter plot showing log2 fold changes of HA-immunoprecipitated peptides from ESC + H/F- Pramel7$_{\Delta N}$ vs ESC + H/F-PRAMEL7$_{WT}$. PRAMEL7$_{\Delta N}$-interacting proteins downregulated in ESC + H/F-PRAMEL7$_{WT}$ are highlighted. Data are from two independent experiments. (G) Cellular component enrichment analysis of PRAMEL7$_{\Delta N}$-interacting proteins. Proteins downregulated in ESC + H/F-PRAMEL7$_{WT}$ are highlighted in red. Source data are available online for this figure.

Interestingly, one of the factors downregulated at both protein and mRNA levels is *Grb10*, an imprinting gene that is positively regulated by DNA methylation (Hikichi et al, 2003). This result is consistent with previous data showing the role of PRAMEL7 in driving genome hypomethylation through UHRF1 downregulation (Graf et al, 2017). We also found that DNA methyltransferase 1 (DNMT1) levels were reduced in ESCs expressing PRAMEL7. DNMT1 in complex with UHRF1 is responsible for DNA methylation maintenance during DNA replication, suggesting that PRAMEL7, by targeting UHRF1, can also affect DNMT1 stability. SILAC-MS measurements also revealed downregulation of Chromobox Protein Homolog 3 and 5 (CBX3 or Heterochromatin Protein HP1γ and CBX5 or Heterochromatin Protein HP1α) that are involved in the establishment of repressive chromatin (Maison and Almouzni, 2004). Notably, ESC + H/F-PRAMEL7$_{WT}$ cells showed decreased levels of CHD4, MTA3 and RBBP7 that are members of the Nucleosome Remodeling and Deacetylase (NuRD) complex. NuRD acts to maintain ESC identity by controlling the expression of pluripotency and differentiation-associated genes (Kloet et al, 2018; Mor et al, 2018; Reynolds et al, 2012; Zhao et al, 2017). Analysis of STRING database of interaction showed that proteins downregulated by PRAMEL7 have a significant enrichment in pathways involved in chromatin organization and small molecule and/or carbolic acid metabolic processes (Fig. 3C, Dataset EV4). In contrast, PRAMEL7-upregulated proteins were mainly enriched in metabolic processes (Fig. 3D, Dataset EV4). These results suggest that PRAMEL7 might target for CUL2-mediated degradation factors implicated in repressive chromatin organization, including NuRD complex.

## PRAMEL7 directly targets factors implicated in the formation of repressive chromatin that are downregulated in a CUL2-dependent manner

To determine whether proteins downregulated upon PRAMEL7 expression are direct targets of PRAMEL7, we performed two independent HA-IPs followed by mass-spectrometry analyses in ESC + H/F-PRAMEL7$_{WT}$ and parental ESCs (Fig. 3E, Dataset EV5). To identify PRAMEL7-interacting proteins, we used as selection criteria when in both experiments the HA-immunoprecipitated peptides from ESC + H/F-PRAMEL7$_{WT}$ were ≥ 2 peptides and had ≥log$_2$ 0.5-fold peptide number in IPs of ESC + H/F-PRAMEL7$_{WT}$ relative to IPs of parental ESCs. Finally, we defined PRAMEL7-interacting proteins when their average log$_2$ fold enrichment over parental ESC was ≥1. Using these criteria, we identified 169 PRAMEL7-interacting proteins that are linked to several processes, including metabolism and gene expression (Fig. 3E, Dataset EV5). As expected, we also identified CUL2 as a PRAMEL7-intercating protein. Surprisingly, we could not detect UHRF1 as PRAMEL7-interacting

protein. However, UHRF1 is a known target of PRAMEL7 and its association with PRAMEL7 has been experimentally validated in previous work using IP followed by western blot (Graf et al, 2017). Thus, we reasoned that the detection of UHRF1, and maybe of other PRAMEL7-interacting proteins, could be hampered by the fact that proteins interacting with PRAMEL7 are targeted for degradation, leading to a low enrichment of the corresponding peptides in PRAMEL7$_{WT}$-immunoprecipitates. To test this, we performed HA-IPs in ESC + H/F-PRAMEL7$_{\Delta N}$ since we reasoned that the lack of interaction with CUL2 and consequent proteasome degradation could allow the detection of proteins associating with PRAMEL7 (Fig. 3F). Using the criteria described above, we compared the number HA-immunoprecipitated peptides from ESC + H/F-PRAMEL7$_{WT}$ and ESC + H/F-PRAMEL7$_{\Delta N}$ relative to parental ESCs. We found that the large majority of proteins (261) were more enriched in PRAMEL7$_{\Delta N}$-immunoprecipitates compared to PRAMEL7$_{WT}$ while only 10 proteins were more enriched in PRAMEL7$_{WT}$ samples. 92% of the proteins enriched in PRAMEL7$_{\Delta N}$-immunoprecipitates (241) could not be detected in the PRAMEL7$_{WT}$ IPs, suggesting that these are PRAMEL7-targets for CUL2-mediated proteasomal degradation and can only be detected when PRAMEL7-CUL2 interaction is abrogated. Consistent with this, we could detect the association of UHRF1 with PRAMEL7$_{\Delta N}$ (Fig. 3F, Dataset EV6). STRING cellular component analysis of proteins enriched in PRAMEL7$_{\Delta N}$-IPs revealed a strong enrichment of chromatin factors (log$_{10}$ FDR 10$^{-18}$) linked to heterochromatin (log$_{10}$ FDR 10$^{-5}$) (Fig. 3G, Dataset EV6). Further, we found that DNMT1 was also significantly enriched, as well as several members of the NuRD complex (CHD4, RBBP7, MTA2, CSNK2A1, HDAC1), histone deacetylase complex (RUVBL1, RBBP7, CHD4, MTA2, CSNK2A1, HDAC, RUVBL2, SIN3A), and ESC/E(Z) complex (RBBP7, EZH2, JARID2). We also found that PRAMEL7$_{\Delta N}$ associates with MSH2 and MSH6 that form MutSα, a key complex for DNA mismatch repair that was rreported to associate with the DNA methylation machinery and recruited to post-replicative DNA in ESCs (Wang et al, 2016). Importantly, several of these factors were also found to be significantly downregulated on the protein levels in ESC + H/F-PRAMEL7$_{WT}$ in SILAC-MS experiments (Fig. 3A,F). These results indicate that PRAMEL7 directly targets factors implicated in the formation of repressive chromatin that are downregulated in a CUL2-dependent manner.

## PRAMEL7 recruits CUL2 to chromatin

To determine how PRAMEL7 and CUL2 reprogram ESC+serum toward ground-state pluripotency, we assessed the localisation of PRAMEL7$_{WT}$ and PRAMEL7$_{\Delta N}$ by chromatin fractionation (Fig. 4A). We found that the large majority (>80%) of PRA-MEL7$_{WT}$ binds to chromatin (Fig. 4A,B). In contrast, about 60% of

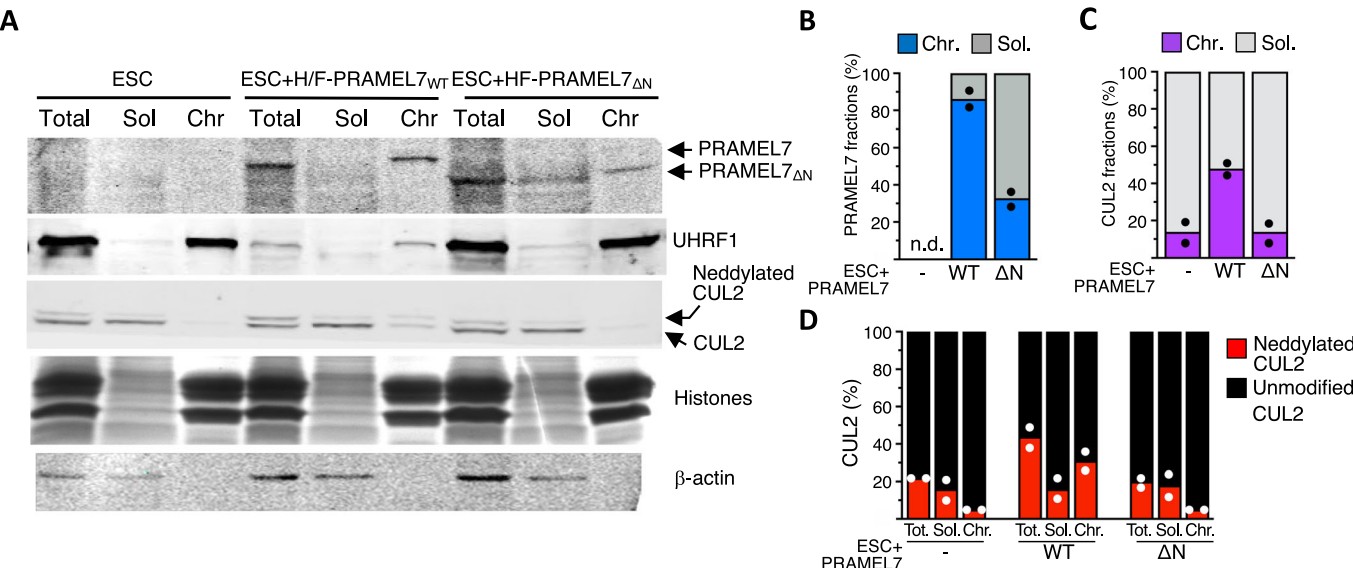

**Figure 4. PRAMEL7 recruits CUL2 to chromatin.**

(A) Cell fractionation of ESC, ESC + H/F-PRAMEL7$_{WT}$, and ESC + H/F-PRAMEL7$_{\Delta N}$. Chromatin-bound (Chr.) and soluble (Sol.) fractions of equivalent cell numbers were analyzed by Western blot to assess the localization of PRAMEL7$_{WT}$, PRAMEL7$_{\Delta N}$, UHRF1, and CUL2. β-actin and histones are shown as loading and fractionation control. (B–D) Quantifications of PRAMEL7$_{WT}$, PRAMEL7$_{\Delta N}$, and total and neddylated CUL2 distribution in total (Tot.), chromatin (Chr.), and soluble (Sol.) fractions in ESC, ESC + H/F-PRAMEL7$_{WT}$, and ESC + H/F-PRAMEL7$_{\Delta N}$. Mean values from two independent experiments are shown. Source data are available online for this figure.

PRAMEL7$_{\Delta N}$ was present in the soluble fraction, indicating that the N-terminal domain of PRAMEL7 is required for the interaction with chromatin. As discussed later in more details, despite the binding of PRAMEL7 to chromatin, we have never been able to obtain specific signals in PRAMEL7-ChIPseq, suggesting that PRAMEL7 does not directly interact with DNA or histones. Next, we examined whether the expression of PRAMEL7 in ESCs could affect CUL2 localization. In parental ESCs and ESC + H/F-PRAMEL7$_{\Delta N}$, CUL2 was predominantly found in the soluble fraction (>90%) whereas in ESC + H/F-PRAMEL7$_{WT}$ about 48% of CUL2 was associated with chromatin (Fig. 4A,C). Notably, about one third (31%) of chromatin-bound CUL2 in ESC + H/F-PRAMEL7$_{WT}$ corresponded to neddylated CUL2 whereas CUL2 in the soluble fraction was mainly unmodified (Fig. 4D). Moreover, chromatin bound CUL2 in parental ESCs and cells expressing Pramel7$_{\Delta N}$ were also unmodified. These results indicate that PRAMEL7 tethers CUL2 to chromatin where it is also enriched in its neddylated, active form. Further, they suggest that the initial steps of PRAMEL7 targeting for proteasomal degradation through CUL2 might occur on chromatin.

To determine which proteins associate with CUL2 on chromatin, we isolated chromatin from parental ESCs and ESC + H/F-PRAMEL7$_{WT}$ and performed CUL2-chromatin-IP followed by mass-spec measurements (Fig. 5A, Dataset EV7). Since the expression of PRAMEL7 promotes CUL2 association with ESC chromatin, the identification of CUL2-interacting proteins on chromatin should also reflect the association of PRAMEL7 with chromatin-bound proteins. We performed two independent experiments and found 547 proteins associating with CUL2 on chromatin of ESC + H/F-PRAMEL7$_{WT}$ compared to parental ESCs (fold changes ≥ 3). We intersected the list of PRAMEL7-interacting

proteins defined by their association with PRAMEL7$_{\Delta N}$ and found that 96 of them (37%) are associated with CUL2 on chromatin of ESC + H/F-PRAMEL7$_{WT}$ (Fig. 5B, Dataset EV7). Interestingly, one of the top pathways of these PRAMEL7- and CUL2-interacting proteins was mechanisms associated with pluripotency (log$_{10}$ FDR 10$^{-11}$) (Dataset EV7). Further, we found that 13 of these PRAMEL7 and CUL2 interacting proteins were also detected as downregulated in ESC + H/F-PRAMEL7$_{WT}$ in SILAC-MS experiments (ALDH2, CHD4, DNMT1, MCM4, MSH2, MSH6, NOP2, NUMA1, RBBP7, SND1TDH, UBA1, UHRF1). We also found a significant enrichment in cellular component linked to heterochromatin (Fig. 5C). These results indicate that PRAMEL7 and CUL2 associate on chromatin with factors implicated in the formation of repressive chromatin, whose stability depends on PRAMEL7 and CUL2.

## PRAMEL7-CUL2 axis antagonizes the repression of genes associated with NuRD complex

The results above showed the association of PRAMEL7 and CUL2 and downregulation in ESC + H/F-PRAMEL7$_{WT}$ of several components of NuRD complex, including CHD4 (Torchy et al, 2015), which was one of the top hit among proteins interacting with CUL2 on chromatin in ESC expressing PRAMEL7 (Fig. 5A). We validated CHD4 interaction with CUL2 and PRAMEL7 by performing CHD4-IP of chromatin of ESC + PRAMEL7 (Fig. 6A). As expected, in the absence of PRAMEL7 expression, we did not detect any CHD4 interaction with CUL2, supporting the role of PRAMEL7 in mediating CHD4-CUL2 interaction. The NuRD complex is a repressive chromatin remodeling complex that has been shown to drive developmental transitions of pluripotent cells and lineage commitment in embryos and ESCs (dos Santos et al, 2014; Kaji et al, 2007; Reynolds et al, 2012; Zhao et al, 2017).

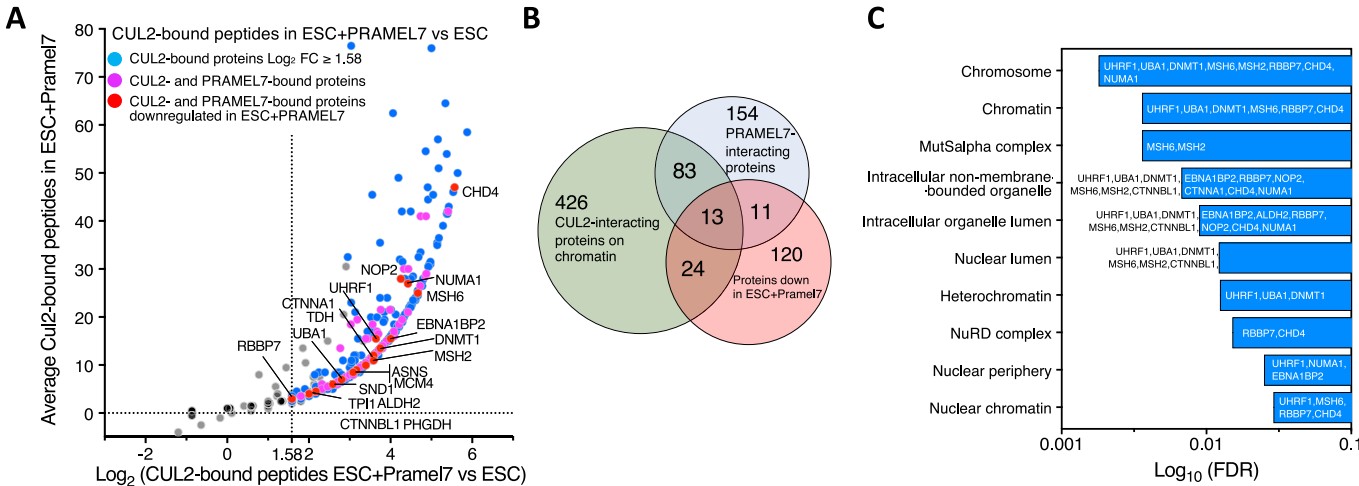

**Figure 5. CUL2 associates with CHD4 on chromatin.**

(A) Scatter plot showing log₂ fold changes of CUL2-immunoprecipitated peptides from chromatin of ESC + H/F-PRAMEL7$_{\Delta N}$ vs ESCs. CUL2- and PRAMEL7-interacting proteins downregulated in ESC + H/F-PRAMEL7$_{WT}$ are highlighted. Data are from two independent experiments. (B) Venn diagrams showing the intersection between CUL2-interacting proteins on chromatin, PRAMEL7-interacting proteins, which correspond to protein identified as PRAMEL7$_{\Delta N}$-interacting proteins, and proteins downregulated in ESC + H/F-PRAMEL7$_{WT}$. (C) Cellular component enrichment analysis of CUL2- and PRAMEL7-interacting proteins. Source data are available online for this figure.

These results prompted us to determine whether PRAMEL7-mediated reprogramming of ESC+serum toward ground-state pluripotency gene expression states involves the regulation of NuRD complex through PRAMEL7-CUL2 axis. Consistent with the SILAC experiments (Fig. 3A), the reduction of global CHD4 levels upon PRAMEL7 expression is low and undetectable by western blot analysis (Fig. 6A), suggesting that the effect on CHD4 protein levels mediated by PRAMEL7-CUL2 axis might occur only to a minor fraction of it. We searched for PRAMEL7-regulated genes that are bound by the NuRD complex by intersecting the RNAseq of ESC + H/F-PRAMEL7$_{WT}$ (Fig. 2) and published CHD4-ChIPseq performed in ESC+serum (Kloet et al, 2018). We found that CHD4 associates with the promoter of 229 PRAMEL7-regulated genes (Fig. 6B, Dataset EV8). Importantly, 73% of these genes (166) were PRAMEL7$_{Cul2}$-genes and a large fraction of them (70%) were upregulated in ESCs expressing PRAMEL7 compared to parental ESCs (Fig. 6C,D, Dataset EV8). KEGG analysis revealed that upregulated PRAMEL7$_{Cul2}$-genes with CHD4-bound promoter are significantly enriched in signaling pathways, including the regulation of pluripotency in stem cells (Fig. 6E, Dataset EV8). Accordingly, the promoter of these genes showed a significant enrichment in the pluripotency factor SOX2 binding motifs (Fig. 6F, Dataset EV8). These results suggest that PRAMEL7-CUL2 axis increases the expression of a set of NuRD-bound genes, thereby contrasting the repressive effect of NuRD. To test this, we performed CHD4-ChIP and compared CHD4 occupancy between parental ESCs, ESC + H/F-PRAMEL7$_{WT}$, and ESC + H/F-PRAMEL7$_{\Delta N}$ (Fig. 6G,H, Appendix Fig. S1A,B). We found that the association of CHD4 at gene promoters of ESCs expressing PRAMEL7 was drastically reduced compared to control ESCs and ESCs expressing PRAMEL7$_{\Delta N}$. We validated these results by measuring CHD4-occupancy at three upregulated PRAMEL7$_{Cul2}$-genes bound by CHD4 (*Bcl3*, *Tgif2*, and *Esrra*) between parental ESCs, ESC + H/F-PRAMEL7$_{WT}$, and ESC + H/F-PRAMEL7$_{\Delta N}$ (Fig. 6I). Only few genomic sites gained CHD4 signal

in ESC + H/F-PRAMEL7$_{WT}$, indicating that the expression of PRAMEL7 did not cause a relocalization of CHD4 but instead promoted a loss of a stable association of CHD4 with chromatin. Accordingly, the few promoters (100) which become bound by CHD4 in ESC + PRAMEL7 showed a weak CHD4 signal compared to average CHD4 signal in control cells and in general they were not significantly downregulated (only two genes were downregulated) (Appendix Fig. S1C). Consistent with the ChIPseq data, a large fraction (69%) of genes that lost CHD4 binding in ESC + H/F-PRAMEL7$_{WT}$ and were regulated by PRAMEL7 correspond to PRAMEL7$_{Cul2}$-genes and a large portion of them (72%) were upregulated in ESCs expressing PRAMEL7 compared to parental ESCs (Fig. 6J,K). Finally, to determine whether PRAMEL7 expression can affect NuRD-regulated genes, we used published RNAseq data of ESC+serum depleted of MBD3, a component of NuRD that maintains its structural integrity and whose genome occupancy is almost completely coincident with CHD4 (Bornelov et al, 2018; Luo et al, 2015; Zhang et al, 2016). We found that about 32% of genes upregulated in ESC + PRAMEL7 were also significantly regulated upon *Mbd3*-KD and 62% of them (179 genes) were upregulated (Appendix Fig. S1D, Dataset EV1). These results further support a role of PRAMEL7 in regulating NuRD activity in ESCs.

All together, these results implicate PRAMEL7-CUL2 axis in the regulation of NuRD complex. This process could occur either by causing site-specific degradation of NuRD complex or by displacing it from chromatin with consequential increase in the expression of target genes.

## Discussion

ESCs need to be fed by extrinsic signals to indefinitely retain pluripotency in culture. In this work, we show that PRAMEL7 in

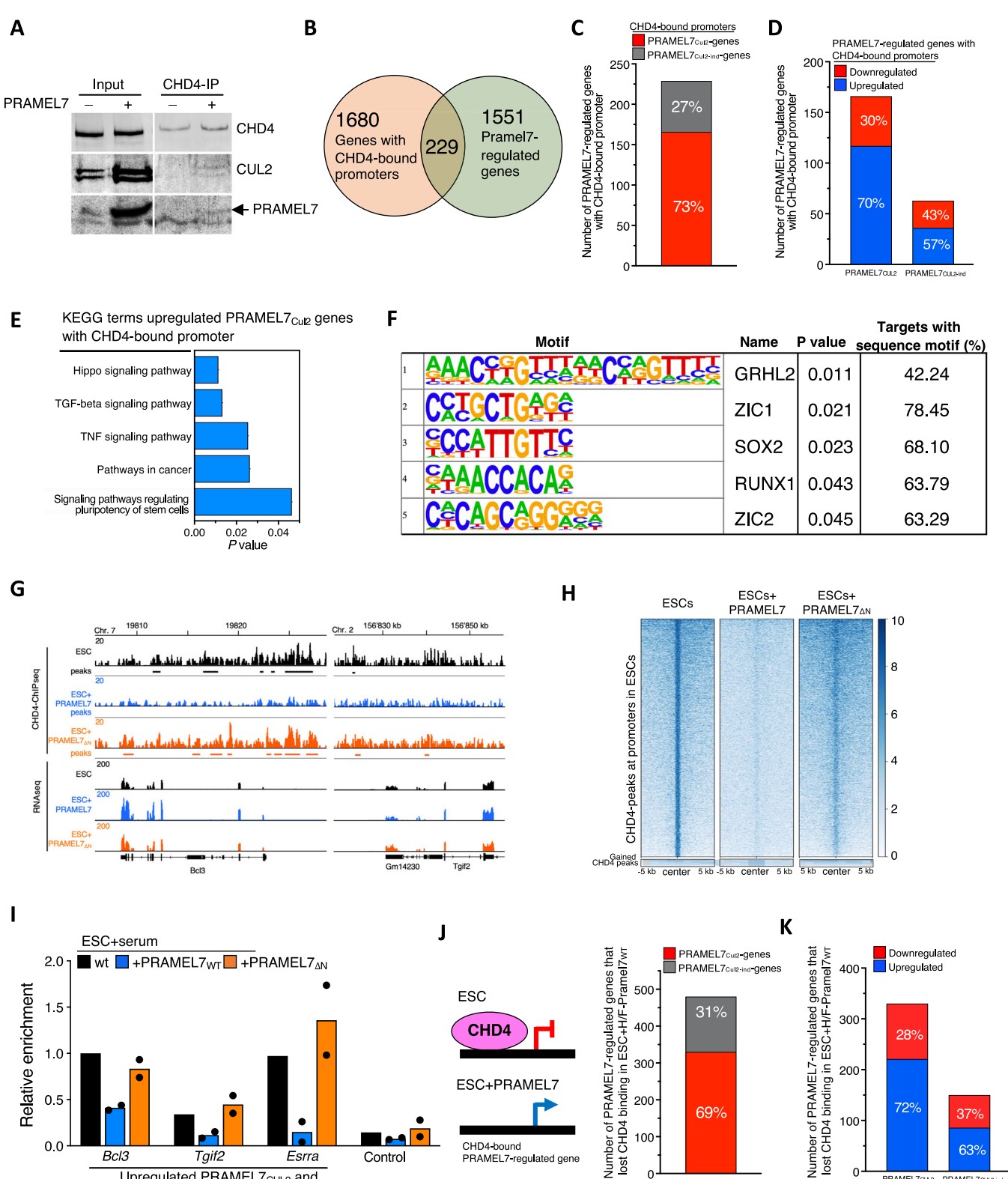

combination with CUL2 reprograms developmentally advanced ESC+serum toward a ground-state gene expression signature by affecting the stability of chromatin repressors. Previous work showed that increasing the expression of PRAMEL7 in

ESC+serum, which express PRAMEL7 at low levels compared to ICM, induced proteasomal degradation of UHRF1, thereby leading to DNA hypomethylation similarly to ICMs (Graf et al, 2017). However, DNA hypomethylation alone cannot entirely explain the

◄ **Figure 6. PRAMEL7-CUL2 axis contrasts the repression of genes associated with NuRD complex.**

(A) CHD4-immunoprecipitation (IP) in parental ESCs and ESC + H/F-PRAMEL7. Western blot shows CHD4, CUL2, and PRAMEL7. (B) Venn diagram showing the intersection of genes with CHD4-bound promoters and PRAMEL7-regulated genes. (C) Number of PRAMEL7$_{Cul2}$- and PRAMEL7$_{Cul2-ind}$-genes with CHD4-bound promoters obtained from published CHD4-ChIPseq data (Kloet et al, 2018). (D) Number of upregulated and downregulated Pramel7$_{Cul2}$- and Pramel7$_{Cul2-ind}$-genes with CHD4-bound promoters. (E) KEGG Pathway enrichment analysis of upregulated PRAMEL7$_{Cul2}$- genes with CHD4-bound promoters using the DAVID annotation tool. (F) Transcription factor motif of the promoters of upregulated PRAMEL7$_{Cul2}$-genes with CHD4-bound promoters by Homer analysis. (G) Representative images showing CHD4-association in ESCs (ChIPseq) and gene expression (RNAseq) in parental ESCs, ESC + H/F-PRAMEL7$_{WT}$, and ESC + H/F-PRAMEL7$_{\Delta N}$ of two upregulated PRAMEL7$_{Cul2}$- genes with CHD4-bound promoters. The corresponding input signals can be found in Appendix Fig. S1A. (H) Heatmap showing CHD4 peaks at promoters detected in parental ESCs and the corresponding signals in ESC + H/F-PRAMEL7$_{WT}$ and ESC + H/F-PRAMEL7$_{\Delta N}$. (I) CHD4 ChIP-qPCR in parental ESCs, ESC + H/F-PRAMEL7$_{WT}$, and ESC + H/F-PRAMEL7$_{\Delta N}$ of three upregulated PRAMEL7$_{Cul2}$-genes with CHD4-bound promoters (Esrra, Bcl3, Tgif2). The control represents a sequence not bound by CHD4. Values were normalized to input and CHD4 occupancy to Bcl3 gene in parental ESCs. Mean values from two independent experiments are shown. (J) Number of PRAMEL7$_{Cul2}$- and PRAMEL7$_{Cul2-ind}$-regulated genes that lost CHD4 binding in ESC + H/F-PRAMEL7$_{WT}$. (K) Number of upregulated and downregulated PRAMEL7$_{Cul2}$- and PRAMEL7$_{Cul2-ind}$-genes that lost CHD4 binding in ESC + H/F-PRAMEL7$_{WT}$. Source data are available online for this figure.

transcriptional changes occurring upon expression of PRAMEL7 in ESC+serum since loss DNA methylation in ESCs does not lead to significant changes in gene expression (Habibi et al, 2013). Further, *Uhrf1* and *Dnmt1* null embryos undergo embryonic lethality during early- to mid-gestation (Li et al, 1992; Sharif et al, 2007) whereas *Pramel7* null embryos arrest much earlier in development, at morula stage, suggesting that PRAMEL7 function in embryos is not only limited to the regulation of DNA methylation.

Here, we have shown that the role of PRAMEL7 in establishing a ground-state gene signature depends on its interaction with CUL2 (Fig. 7). PRAMEL7/CUL2 axis affects the stability of several chromatin factors that are linked to repressive chromatin. The identification PRAMEL7/CUL2 targets that are components of DNA methylation maintenance machinery (DNMT1 and UHRF1 and MutSα complex) support the role of PRAMEL7 in regulating the stability of the DNA methylation machinery in ESCs through proteasome degradation and, consequently, global DNA hypomethylation (Graf et al, 2017). We have also found several other chromatin repressors whose protein stability is regulated by PRAMEL7-CUL2 axis and can associate with PRAMEL7 and CUL2. The PRAMEL7-CUL2 regulation of these factors can contribute to the establishment of ground-state gene expression, which is characterized by a low repressive chromatin state. Among these proteins, we found several members of the NuRD complex, including CHD4. Accordingly, a large majority of PRAMEL7-regulated genes with CHD4-bound promoters are genes whose upregulation in ESCs expressing PRAMEL7 depends on PRAMEL7-CUL2 interaction (Pramel7$_{Cul2}$ genes). Although a small, but significant decrease of CHD4 mediated by PRAMEL7-CUL2, we observed a strong decrease of CHD4 genome occupancy by ChIPseq in ESC + PRAMEL7 but not in ESC + PRAMEL7$_{\Delta N}$. These results suggest that PRAMEL7-CUL2 axis might operate in two distinct ways, either by targeting a small fraction of NuRD components for degradation or by displacing NuRD from chromatin, thereby decreasing the repression of genes that can be implicated in ground-state pluripotency. Consistent with these results, upregulated PRAMEL7$_{CUL2}$ genes with CHD4-bound promoters were enriched in pathways linked to pluripotency and in the pluripotency transcription factor SOX2 motif. Accordingly, studies in ESC-KO for *Mbd3*, a factor essential for proper assembly of the NuRD complex, showed that several genes linked to pluripotency are repressed by NuRD (Reynolds et al, 2012). Our results proposed PRAMEL7 as one of the factors that can antagonise NuRD function for the establishment of PRAMEL7-

mediated ground-state pluripotency. It was reported that embryos lacking CHD4 can form a morphologically normal morula but are unable to form functional trophectoderm due to *Nanog* upregulation (O'Shaughnessy-Kirwan et al, 2015). Consequentially, in the absence of a functional trophectoderm, *Chd4*-KO blastocysts do not implant and are hence not viable. The expression of PRAMEL7 in both the morula and ICM might suggest a function in regulating CHD4 activity specifically in these early developmental stages. Although our results have highlighted a role of PRAMEL7-CUL2 in the regulation of NuRD, our data do not exclude that the other identified chromatin repressors targeted by PRAMEL7-CUL2 could also contribute to the reprogramming into ground-state gene expression state.

We have shown that the expression of PRAMEL7 promotes CUL2 recruitment to chromatin, suggesting that the initial steps for targeting to proteasomal degradation are on chromatin. This recruitment is directly mediated by PRAMEL7 since the expression PRAMEL7$_{\Delta N}$-mutant does not promote CUL2 association with chromatin. CUL2 recruitment to chromatin has also been reported for COMMD1, an inhibitor of NF-κB that interacts and recruits CUL2 to chromatin to accelerate the ubiquitination and degradation of NF-κB subunits in human cells (Maine et al, 2007). Interestingly, we found that CUL2 bound to chromatin of ESCs expressing PRAMEL7 is enriched in CUL2 modified with a NEDD8 adduct that increases its ubiquitination activity and is required for CUL2 activation (Hori et al, 1999; Ohh et al, 2002). Studies on *Cul2*-KO embryos have not yet been reported. However, embryos KO for the *Uba3* gene, which expresses a catalytic subunit of NEDD8-activating enzyme (NAE), have shown selective apoptosis of the ICM, indicating an essential role of NEDD8 in concert with Cullin family proteins during preimplantation development, the time when also PRAMEL7 is expressed and required for development (Graf et al, 2017; Tateishi et al, 2001; Wada et al, 1999).

We have also observed that PRAMEL7$_{\Delta N}$-mutant showed reduced association with chromatin, indicating a role of its N-terminal domain in this process. However, it remains yet to be elucidated how PRAMEL7 associates with chromatin. Despite all our attempts to obtain PRAMEL7-ChIPseq, we have never been able to obtain specific signals, suggesting that PRAMEL7 does not directly interact with DNA or histones, as also evident by the lack of DNA- or histone-binding domains. Future work will be addressed to determine how PRAMEL7 is tethered to specific chromatin loci.

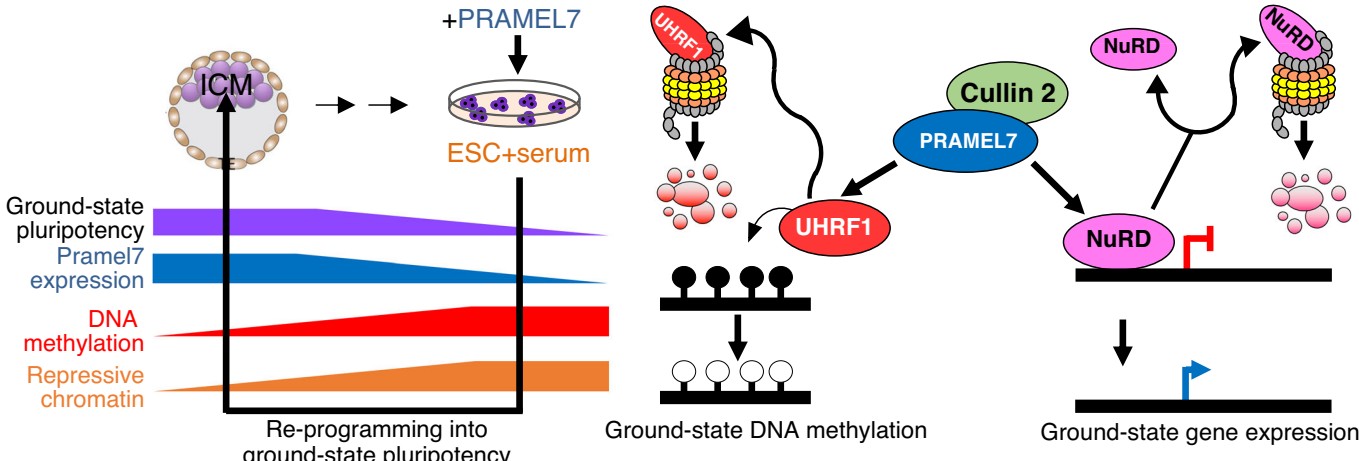

**Figure 7. Model showing how PRAMEL7 reprograms ESCs into a ground-state epigenetic and gene expression signature.**

Expression levels of PRAMEL7 correlate with ground-state pluripotency, high in ICM and low in ESC+serum. They also correlate with a reduction of repressive chromatin. PRAMEL7 association with CUL2 targets UHRF1 for proteasomal degradation and induces a global loss of DNA methylation, resembling the DNA hypomethylation state of the ICM (Graf et al, 2017). On the other hand, PRAMEL7 recruits CUL2 to chromatin and targets chromatin repressors for proteasomal degradation, including the NuRD complex. In the schema is shown how PRAMEL7-CUL2 can derepress NuRD-regulated genes either by displacing NuRD from chromatin or causing degradation of NuRD with consequential increase in the expression of target genes that can have an impact in the establishment of ground-state gene expression signature.

In summary, we showed that PRAMEL7 targets the stability of chromatin repressive factors in a CUL2 dependent manner, thereby promoting a ground-state gene expression signature. The identification of this process revealed an as-yet-unappreciated link between proteasome pathways and chromatin regulation for the establishment of pluripotency and might offer potential strategies for the optimization of methodologies to reprogram cells back to ground-state pluripotency.

# Methods

## Cell culture

129 mouse embryonic stem cells (E14 line) were cultured in serum medium containing Dulbecco's modified Eagle's medium + Glutamax (Life Technologies), 15% FCS (Life Technologies; Cat no. 10270106 FBS South American), 1× MEM NEAA (Life Technologies), 100 μM β-mercaptoethanol, recombinant leukemia inhibitory factor, LIF (Polygene, 1000 U/ml), 1× penicillin/streptomycin (Life Technologies). ESCs were seeded at a density of 50,000 cells/cm² in culture dishes (Corning® Cell BIND® surface) treated with 0.1% gelatin without feeder layer. Propagation of cells was carried out every 2–3 days using enzymatic cell dissociation.

## Establishment of ESC lines expressing PRAMEL7

ESCs were co-transfected with a plasmid expressing Cas9 protein and the sgRNA guide sequence targeting the *Rosa26* locus (Genome CRISPR™ mouse ROSA26 safe harbor gene knock-in kit, SH054, GeneCopoeia) and the HDR repair template plasmid containing HA-FLAG-Pramel7$_{WT}$ and -Pramel7$_{\Delta N}$ sequences under the CAG promoter, flanked by the homology arms (molar ratio 1:3). Two days after transfection, ESCs were selected using 2 μg of Puromycin

(Life Technologies) overnight. After recover, ESCs were further treated with 1 μg of Puromycin (Life Technologies) for additional three days. After three days of culture, cells were seeded for single cell clone isolation. Puromycin resistant ESC clones were genotyped using PCR primers that could distinguish between insertions of the construct in one or both alleles (Dataset EV9).

## RNA sequencing and data analysis

$2.5 \times 10^6$ cells from ESCs, ESC + H/F-PRAMEL7$_{WT}$, and ESC + H/F-Pramel7$_{\Delta N}$ were collected in triplicates, and the total RNA was purified with TRIzol reagent (Life Technologies). In order to remove DNA contaminants, samples were treated with 1U DNase I (Thermo Scientific) for 1 h at 37 °C and RNA samples were re-purified using TRIzol and Chloroform extraction. The RNA samples (100–1000 ng) were polyA enriched and then reverse-transcribed into double-stranded cDNA. The cDNA samples were fragmented, end-repaired, and adenylated before ligation of TruSeq adapters containing unique dual indices (UDI) for multiplexing. Fragments containing TruSeq adapters on both ends were selectively enriched with PCR. The quality and quantity of the enriched libraries were validated using Qubit® (1.0) Fluorometer and the TapeStation 2200 (Agilent Technologies Inc., Santa Clara, California). The product is a smear with an average fragment size of approximately 260 bp. Libraries were normalized to 10 nM in Tris-Cl 10 mM, pH8.5 with 0.1% Tween 20. The Novaseq 6000 (Illumina, Inc, California, USA) was used for cluster generation and sequencing according to standard protocol. Sequencing was single end 100 bp. The quality of the generated 120 bp single end reads was checked by FastQC, a quality control tool for high throughput sequence data (Andrews, 2010). The quality of the reads was increased by applying: (a) SortMeRNA (version 2.1) tool to filter ribosomal RNA (Kopylova and Touzet 2012); (b) Trimmomatic (version 0.36) software package to trim the sorted

(a) reads (Bolger et al, 2014). The sorted (a), trimmed (b) reads were mapped against the mouse genome (mm10) using the default parameters of the STAR (Spliced Transcripts Alignment to a Reference, version 2.4.0.1) (Dobin et al, 2013). For each gene, exon coverage was calculated using a custom pipeline and then normalized in reads per kilobase per million (RPKM) (Mortazavi et al, 2008), the method of quantifying gene expression from RNA sequencing data by normalizing for total read length and the number of sequencing reads. Only genes with more than one read per million in at least two samples were conserved for the analysis. Genes were classified as differentially expressed with an adjusted $P$-value (FDR) below 0.05 for DESeq2 and a minimal logFC of 1.5 for EdgeR. The functional annotation analysis of the differentially expressed genes was performed using DAVID (Database for Annotation, Visualization and Integrated Discovery) (Huang da et al, 2009). RNAseq data of *Mbd3*-KD ESCs are from GEO accession number GSE61188 (Luo et al, 2015).

## Principal component analysis

The PCA has been performed on qRT-PCR data set combined with the RNAseq data described in (Boroviak et al, 2014). The $\log_2$ counts in RNAseq have been selected and normalized to mean. The data set taken for PCA and hierarchical clustering was created as a projection of 96 genes used in (Boroviak et al, 2014). onto the PRAMEL7 ESCs RNAseq experiment. The data was RPKM normalized and scaled prior to PCA removing the lower 5% of least variance. PCA was performed using PCAtools function in R package. (PCAtools: PCAtools: Everything Principal Components Analysis. R package version 2.6.0, https://github.com/kevinblighe/PCAtools).

## Cell fractionation

ESCs were collected by trypsinization, washed once with PBS, and counted. ESC pellets were resuspended at a concentration of $20 \times 10^6$ cells/ml PBS, divided into two, and centrifuged at $1000 \times g$ for 5 min). One sample was used for whole cell extract and the other for the chromatin fractionation. For the whole cell extract, cell pellets were resuspended in MNase Buffer containing 0.3 M Sucrose, 50 mM Tris-HCl pH 7.5, 30 mM KCl, 7.5 mM NaCl, 4 mM MgCl2, 1 mM CaCl2, 0.125% NP-40, fresh 0.25% NaDeoxycholate, and 1x cOmplete™ Protease Inhibitor Cocktail (Roche). Samples were incubated with MNase (S7 Micrococcal nuclease, Roche) in the ratio 5 Units MNase/1 million cells at 37 °C for 30– 45 min and boiled in 1x Laemmli buffer (10% glycerol, 10 mM Tris pH 6.8, 2% SDS, 0.1 mg/ml bromophenolblue, 2% β-mercaptoethanol) at 95 °C for 5 min. For the chromatin fractionation, ESC pellets were resuspended with the chromatin extraction buffer (10 mM Hepes pH 7.6, 150 mM NaCl, 3 mM MgCl₂, 0.5% Triton X-100, freshly supplemented with cOmplete™ Protease Inhibitor Cocktail). Sample was then sonicated for 15 s to lyse nuclei and incubated for 30 min at room temperature rotating. Precipitated chromatin was isolated by centrifugation at $3000 \times g$ for 10 min at 4 °C. The soluble fraction (supernatant) was collected into a new tube and boiled in 1x Laemmli buffer at 95 °C for 5 min. Chromatin pellet was incubated with MNase (S7 Micrococcal nuclease, Roche) to ensure genomic DNA fragmentation. Chromatin fractions were then boiled in 1x Laemmli buffer at 95 °C for 5 min and further analyzed by Western Blotting.

## Stable isotope labeling by amino acids (SILAC)

ESCs, ESC + H/F-PRAMEL7$_{WT}$, and ESC + H/F-PRAMEL7$_{\Delta N}$ were cultured in DMEM media for SILAC (ThermoFischer Scientific; 89985) with 15% knockout serum replacement (ThermoFischer Scientific; 10828010, substituted with 1× MEM NEAA (Life Technologies), 100 µM β-mercaptoethanol, recombinant leukemia inhibitory factor, LIF (Polygene, 1000 U/ml)). Cells were cultured in the above media containing either 0.15 mg/ml Lys-8 HCl L-Lysine HCl 13C,15N (Silantes; 211603902) replacing normal Lys and 0.09 mg/ml Arg-10 HCl L-Arginine HCl 13C,15N (Silantes; 201603902) replacing normal Arg or in media containing 0.15 mg/ml L-Lysine monohydrochloride (Sigma; L5626) as a Lys replacement and 0.09 mg/ml L-arginine monohydrochloride (Sigma; A6969) as a Arg replacement. Before collection for downstream analyses, cells were passaged 4× in the indicated culture media to achieve >95% protein labeling. Cells were washed 2× in PBS, lysed in hot lysis buffer (6 M Guanidine hydrochloride, 5 mM TCEP, 10 mM CAA, 100 mM Tris HCl pH 8). Samples were then sonicated, incubated at 95 °C for 10 min and cleared via centrifuged at $15,000 \times g$ for 10 min. The lysates were then stored at −80 °C until LC-MS/MS analysis.

In preparation for trypsin digestion, 25 µg of light or heavy label ESC+serum lysates (reference quantification lysates) were mixed with 25 µg of the opposing labeled ESC+serum, ESC + H/F-PRAMEL7$_{WT}$ and ESC + H/F-PRAMEL7$_{\Delta N}$ lysates. The 50 µg of proteins were then trypsin digested overnight at 37 °C (1:25; Promega) using the filter aided sample preparation (FASP) methodology (Ostasiewicz et al, 2010). The samples were then acidified with TFA and salts removed using ZipTip C18 pipette tips (Millipore Corp.). The peptides were eluted with 20 µl of 60% ACN, 0.1% TFA, dried to completion and then reconstituted in MS buffer (3% ACN, 0.1% formic acid).

LC-MS/MS analyses were performed on an Orbitrap Fusion Tribrid mass spectrometer (Thermo Fisher Scientific), coupled to a nano EasyLC 1000 liquid chromatograph (Thermo Fisher Scientific). Peptides were loaded on a commercial MZ Symmetry C18 Trap Column (100 Å, 5 µm, 180 µm × 20 mm, Waters) followed by nanoEase MZ C18 HSS T3 Column (100 Å, 1.8 µm, 75 µm × 250 mm, Waters). Peptides were eluted over 100 min at a flow rate of 300 nL/min. An elution gradient protocol from 2 to 25% B, followed by two steps at 35% B for 5 min and at 95% B for 5 min, respectively, was used. The mass spectrometer was operated in data-dependent mode (DDA) acquiring a full-scan MS spectrum ($300 - 1800$ $m/z$) at a resolution of 120,000 at 200 $m/z$ after accumulation to a target value of 500,000. Data-dependent MS/MS were recorded in the linear ion trap using quadrupole isolation with a window of 0.8 Da and HCD fragmentation with 35% fragmentation energy. The ion trap was operated in rapid scan mode with a target value of 10'000 and a maximum injection time of 50 ms. Only precursors with intensities above 5000 were selected for MS/MS and the maximum cycle time was set to 3 s. Charge state screening was enabled. Singly, unassigned, and charge states higher than seven were rejected. Precursor masses previously selected for MS/MS measurement were excluded from further selection for 20 s, and the exclusion window was set at 10 ppm. The samples were acquired using internal lock mass calibration on $m/z$ 371.1012 and 445.1200.

RAW data files were converted to the mzXML format (Pedrioli et al, 2004) and searched against the the Swiss-Prot mouse protein

database version of 2019-08 using version 2018.01 rev. 3 of Comet (Eng et al, 2013). Search parameters used were carboxyamido-methylation (57.021464 Da) of Cys as static modification, 13C6,15N2-Lys (8.01419892 Da), and 13C6,15N4-Arg (10.008252778 Da) as variable modifications, semitryptic digestion with a maximum of two missed cleavages, and 20 ppm error tolerance for precursor ions. Peptide probabilities were evaluated with PeptideProphet (Keller et al, 2002), and ProteinProphet (Nesvizhskii et al, 2003) was used to estimate protein probabilities. Peptides and proteins were filtered for 1% false-discovery rate. Protein abundance ratios were computed as L/H (light/heavy) using XPRESS (Han et al, 2001) with a mass tolerance of 10 ppm and a custom script was used to remove proteins with inconsistent ratios across the SILAC label switch. Significance of differential abundance across conditions was calculated using $t$-test and adjusted for multiple-testing using Benjamini-Hochberg.

## Immunoprecipitation

Immunoprecipitation analysis was performed on ESC+serum, ESC + H/F-PRAMEL7$_{WT}$ and ESC + H/F-PRAMEL7$_{\Delta N}$ cell lines by collecting 30mio cultured cells per cell line. Cells were washed 2× in PBS then incubated with MNase as described above for 30–45 min. Samples were then incubated with 150 mM NaCl for 10 min at 4 °C. Samples were centrifuged at $16,000 \times g$ for 5 min and the supernatant was collected, precleared with 10 μl of packed Sepharose beads for 2 h at 4 °C, and then incubated overnight with Pierce™ Anti-HA Magnetic Beads (Thermo Scientific) at 4 °C. Beads were then washed 3× in C-150 buffer (20 mM Hepes pH 7.6, 20% glycerol, 200 mM NaCl, 1.5 mM MgCl$_2$, 0.2 mM EDTA, 0.02% NP40, and 1x cOmplete™ Protease Inhibitor Cocktail). Purified complexes bound to the beads were submitted for tryptic digestion off the beads and subsequent mass spectrometric analyses.

## Chromatin-IP

Chromatin-IP was performed as previously described (Dalcher et al, 2020). Approximately $10^7$ ESCs were collected by trypsinization, centrifuged, and washed once with PBS. Nuclei were isolated by re-suspending the cells in two consecutive rounds in hypotonic buffer (10 mM Hepes pH 7.6, 1.5 mM MgCl$_2$, 10 mM KCl, 2 mM Na$_3$VO$_4$ freshly supplemented with cOmplete™ Protease Inhibitor Cocktail). The suspension was homogenized using a handheld A pestle 10–20 times and the purity of nuclei was checked under a microscope. The nuclei were then isolated, sonicated for 15 s, and crosslinked in chromatin fractionation (10 mM Hepes pH 7.6, 3 mM MgCl$_2$, 150 mM NaCl, and 0.5% Triton X-100) containing 2 mM Na$_3$VO$_4$ and 0.5 mM dithiobis[succinimidylpropionate] (DSP, Thermo Scientific) and supplemented with cOmplete™ Protease Inhibitor Cocktail. The solution was incubated at room temperature for 30 min by rotation. The crosslinking was quenched by the addition of 25 mM Tris HCl pH 7.5. The chromatin was then isolated by centrifugation and washed twice in MNase buffer. Digestion of chromatin into mononucleosomes was obtained by digestion with 100U MNase in MNase buffer at 37 °C for 45 min. 1% SDS was then added followed by 3× 30 s sonication steps with a bioruptor sonicator (Diagenode) to solubilize the chromatin. Insoluble precipitates were removed by centrifugation and soluble crosslinked chromatin extracts were diluted 10× in IP buffer (0.3 M

Sucrose, 50 mM Tris pH 7.5, 30 mM KCl, 300 mM NaCl, 4 mM MgCl$_2$, 1 mM CaCl$_2$, 0.125% NP-40, 0.25% NaDeoxycholate, 2 mM Na$_3$VO$_4$, supplemented with cOmplete™ Protease Inhibitor Cocktail). Samples were precleared for 2 h with 10 μl of packed Sepharose beads for 2 h at 4 °C. The beads were then centrifuged, and the supernatants were incubated overnight with Cullin2 antibody at a concentration of 4 μg/mL at 4 °C (Dataset EV9). Samples were then incubated with 20 μl equilibrated Dynabeads protein A for 4 h at 4 °C. The beads were then subsequently washed 5× in wash buffer (20 mM Tris pH 7.5, 20% glycerol, 100 mM KCl, 300 mM NaCl, 1.5 mM MgCl$_2$, 0.2 mM EDTA, 0.125% NP40, 0.25% NaDeoxycholate, 2 mM Na$_3$VO$_4$ supplemented with cOmplete™ Protease Inhibitor Cocktail). Purified complexes bound to the beads were submitted for tryptic digestion of the beads and subsequent mass spectrometric analyses.

## Mass spectrometric analysis

The dry beads were washed beads were re-suspended in 45 μl digestion buffer (Tris/2 mM CaCl2, pH 8.2), reduced with 5 mM TCEP (tris(2-carboxyethyl)phosphine) and alkylated with 15 mM iodoacetamide. Proteins were on-bead digested using 5 μl of Sequencing Grade Trypsin (100 ng/μl in 10 mM HCl, Promega). The digestion was carried out in a microwave instrument (Discover System, CEM) for 30 min at 5 W and 60 °C. The supernatants were transferred in new tubes and the beads were washed with 150 μl 0.1% trifluoroacetic acid (TFA) -buffer and combined with the first supernatant. The samples were dried to completeness and re-solubilized in 20 μL of MS sample buffer (3% acetonitrile, 0.1% formic acid).

LC-MS/MS analysis was performed on an Q Exactive mass spectrometer (Thermo Scientific) equipped with a Digital PicoView source (New Objective) and coupled to a nanoAcquity UPLC (Waters Inc.). Solvent composition at the two channels was 0.1% formic acid for channel A and 0.1% formic acid, 99.9% acetonitrile for channel B. Column temperature was 50 °C. For each sample 1 μL of peptides were loaded on a commercial Symmetry C18 trap column (5 μm, 180 μm × 20 mm, Waters Inc.) connected to a BEH300 C18 column (1.7 μm, 75 μm × 150 m, Waters Inc.). The peptides were eluted at a flow rate of 300 nL/min with a gradient from 5 to 35% B in 90 min, 35 to 60% B in 5 min and 60 to 80% B in 10 min before equilibrating back to 5% B.

Samples were measured in randomized order. The mass spectrometer was operated in data-dependent mode (DDA), funnel RF level at 60%, and heated capillary temperature at 275 °C. Full-scan MS spectra (350–1500 $m/z$) were acquired at a resolution of 70,000 at 200 $m/z$ after accumulation to a target value of 3,000,000, followed by HCD (higher-energy collision dissociation) fragmentation on the twelve most intense signals per cycle. Ions were isolated with a 1.2 $m/z$ isolation window and fragmented by higher-energy collisional dissociation (HCD) using a normalized collision energy of 25%. HCD spectra were acquired at a resolution of 35,000 and a maximum injection time of 120 ms. The automatic gain control (AGC) was set to 100,000 ions. Charge state screening was enabled and singly and unassigned charge states were rejected. Only precursors with intensity above 25,000 were selected for MS/MS. Precursor masses previously selected for MS/MS measurement were excluded from further selection for 40 s, and the exclusion window tolerance was set at 10 ppm. The samples were acquired using internal lock mass calibration on m/z 371.1010 and 445.1200.

The mass spectrometry proteomics data were handled using the local laboratory information management system (LIMS) (Türker et al). The acquired raw MS data were converted into Mascot Generic Format files (.mgf) using ProteoWizard (http://proteowizard.sourceforge.net/), and the proteins were identified using the Mascot search engine (Matrix Science, version 2.6.2). Mascot was set up to search the SwissProt database assuming the digestion enzyme trypsin. Spectra were searched against a Uniprot Mus musculus proteome database (taxonomy 10090), concatenated to its reversed decoyed fasta database. Oxidation of methionine was specified in Mascot as a variable modification. Mascot was searched with a fragment ion mass tolerance of 0.030 Da and a parent ion tolerance of 10.0 PPM. Scaffold (Proteome Software Inc., version 5) was used to validate MS/MS-based peptide and protein identifications. Peptide identifications were accepted if they achieved a false discovery rate (FDR) of less than 0.1% by the Scaffold Local FDR algorithm. Protein identifications were accepted if they achieved an FDR of less than 1.0% and contained at least two identified peptides.

### Chromatin immunoprecipitation analysis (ChIP)

ChIP analysis was performed as previously described (Leone et al, 2017). Briefly, 1% formaldehyde was added to cultured cells to cross-link proteins to DNA. Cells were permeabilized with permeabilization buffer containing Triton X-100 (10 mM EDTA pH 8.0, 10 mM EGTA, 10 mM HEPES and 0.25% Triton X-100) and collected. The cell pellet was incubated in MNase buffer, where digestion of chromatin into mononucleosomes was obtained by digestion with 100U MNase in MNase buffer for 30 min at 37 °C. MNase digestion was quenched with 10% EDTA (final volume). 25 µg of chromatin was precleared for 2 h with 10 µl of packed Sepharose beads at 4 °C. The samples were incubated with 2 µg of Chd4 antibody (Dataset EV9) overnight, followed by 4 h with Dynabeads™ Protein G (Thermo Scientific) at 4 °C. The beads were then washed, and the bound chromatin was eluted with the elution buffer (1% SDS, 100 mM NaHCO3). Upon proteinase K digestion (50 °C for 3 h) and reversion of cross-linking (65 °C, overnight), DNA was purified with phenol/chloroform, ethanol precipitated.

ChIP-qPCR measurements were performed with KAPA SYBR® FAST (Sigma) on a Rotor-Gene Q (Qiagen) always comparing enrichments over input samples. Primer sequences are listed in Dataset EV9.

For ChIPseq analyses, the quantity and quality of the isolated DNA was determined with Qubit® 4 Fluorometer (Life Technologies). Libraries were prepared using the NEBNext® Ultra™ II DNA Library Prep for Illumina (New England Biolabs, E7645S and E7645L) following the manufacturer's protocol. Briefly, ChIP and input samples (10 ng) were first end-repaired and polyadenylated. Then, the ligation of Illumina compatible adapters containing the index for multiplexing was performed. The quality and quantity of the enriched libraries were evaluated using Qubit® 4 Fluorometer and 4200 TapeStation System (Agilent). Sequencing was performed on an Illumina NovaSeq6000 machine with single-end 100 bp reads. Motif analysis was performed by using findMotifs.pl from HOMER (v4.11) and applying parameters for motifs of length 8 and 10 from −2 kb to +1 kb bp relative to the transcription start site.

### ChIPseq data analysis

ChIPseq reads were aligned to the mouse mm10 reference genome using Bowtie2 (version 2.3.4.3; (Langmead and Salzberg, 2012)). Read counts were computed and normalized using "bamCoverage" from deepTools (version 3.2.1; (Ramirez et al, 2014)) using a bin size of 50 bp. "computeMatrix" from deepTools was used to generate all heat maps and plot profiles. CHD4 bound regions were defined using SICER (version 1.1; (Zang et al, 2009)) by comparing the CHD4 ChIPs of the three cell lines and the respective input. Integrative Genome Viewer (IGV, version 2.16.0) (Robinson et al, 2011) was used to visualize and extract representative ChIPseq tracks.

## Data availability

RNAseq data generated in this study have been deposited in the NCBI's GEO database under accession code GSE215339. Proteomic data are available via ProteomeXchange with identifier PXD032184.

## Peer review information

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

## Acknowledgements

This work was supported by the Swiss National Science Foundation (31003A_173056 and 201268 to RS), ERC grant (ERC-AdG-787074-NucleolusChromatin to RS), Swiss Cancer Research Foundation (KFS-5488-02-2022 to RS). We thank Paolo Nanni, Sibylle Pfammatter, Simone Wüthrich, Catherine Aquino, and the Functional Genomic Center Zurich for the assistance in sequencing and proteomic analysis. We also thank Dominik Bär for assistance in generating plasmids, Massimiliano Manzo, Luigi Lerra, Rostyslav Kuzyakiv, and Ramon Pfändler for assistance in bioinformatic analyses.

## Author contributions

**Meneka Rupasinghe**: Conceptualization; Data curation; Formal analysis; Investigation; Writing—original draft. **Cristiana Bersaglieri**: Data curation; Formal analysis; Validation; Investigation; Writing—original draft; Writing—review and editing. **Deena M Leslie Pedrioli**: Formal analysis. **Patrick GA Pedrioli**: Formal analysis. **Martina Panatta**: Data curation; Formal analysis. **Michael O Hottiger**: Investigation. **Paolo Cinelli**: Investigation. **Raffaella Santoro**: Conceptualization; Resources; Data curation; Supervision; Funding acquisition; Investigation; Writing—original draft; Project administration; Writing—review and editing.

## Disclosure and competing interests statement

The authors declare no competing interests.

