## [Peer Review File · EMBO Reports]

PRAMEL7 and CUL2 decrease NuRD stability to establish ground-state pluripotency

Meneka Rupasinghe, Cristiana Bersaglieri, Deena Leslie Pedrioli, Patrick Pedrioli, Martina Panatta, Michael Hottiger, Paolo Cinelli, and Raffaella Santoro

DOI: 10.15252/embr.202358165

Corresponding author(s): Raffaella Santoro (raffaella.santoro@dmmd.uzh.ch)

Review Timeline:

Submission Date:	13th Sep 23
Editorial Decision:	10th Oct 23
Revision Received:	4th Dec 23
Editorial Decision:	2nd Jan 24
Revision Received:	5th Jan 24
Accepted:	10th Jan 24

Editor: Esther Schnapp

Transaction Report:

Dear Raffaella,

Thank you for the submission of your manuscript to EMBO reports. We have now received the full set of referee reports that is pasted below.

As you will see, the referees acknowledge that the findings are potentially very interesting. They predominantly have minor concerns, except for one major concern raised by referee 3 that overlaps with a comment by referee 1 and should be addressed, as should all other points. Please let me know in case you disagree and we can discuss the exact revision requirements further, also in a video chat, if you like.

I would thus like to invite you to revise your manuscript with the understanding that the referee concerns must be fully addressed and their suggestions taken on board. Please address all referee concerns in a complete point-by-point response. Acceptance of the manuscript will depend on a positive outcome of a second round of review. It is EMBO reports policy to allow a single round of major revision only and acceptance or rejection of the manuscript will therefore depend on the completeness of your responses included in the next, final version of the manuscript.

We realize that it is difficult to revise to a specific deadline. In the interest of protecting the conceptual advance provided by the work, we recommend a revision within 3 months (10th Jan 2024). Please discuss the revision progress ahead of this time with the editor if you require more time to complete the revisions.

- 1) A data availability section providing access to data deposited in public databases is missing. If you have not deposited any data, please add a sentence to the data availability section that explains that.
- 2) Your manuscript contains statistics and error bars based on $n=2$. Please use scatter blots in these cases. No statistics should be calculated if $n=2$.

5) a complete author checklist, which you can download from our author guidelines <https://www.embopress.org/page/journal/14693178/authorguide>. Please insert information in the checklist that is also

reflected in the manuscript. The completed author checklist will also be part of the RPF.

6) Please note that all corresponding authors are required to supply an ORCID ID for their name upon submission of a revised manuscript (<<https://orcid.org/>>). Please find instructions on how to link your ORCID ID to your account in our manuscript tracking system in our Author guidelines <<https://www.embopress.org/page/journal/14693178/authorguide#authorshipguidelines>>

I look forward to seeing a revised form of your manuscript when it is ready.

Best wishes,
Esther

Referee #1:

This paper by Meneka Rupasinghe and co-authors examines mechanisms of pluripotency in mouse ES cells.

In an excellent previous paper (Graf NCB 2017), the Santoro lab showed that, in the ICM and in 2i conditions, the protein PRAMEL7 is expressed, causes UHRF1 degradation and hypomethylation, and installs an epiblast-like/ground-state transcription program. This paper also established that PRAMEL7 interacts with CUL2, a RING-type E3 ubiquitin ligase.

The current paper builds upon these findings and shows that:

- PRAMEL7 interacts with CUL2 via two consensus boxes in the Nterminus (Figure 1). Deleting these boxes renders PRAMEL7 unable to target UHRF1 for degradation.
- As previously described, expression of PRAMEL7 directs ES cells to have a 2i-like transcriptome, whereas the truncated PRAMEL7 does not have the same effect (Figure 2).
- PRAMEL7 controls the protein abundance of many proteins, including a number of chromatin factors such as DNMT1 and NURD. It interacts directly with some of these complexes, as judged by IP/MS using the DeltaN form (Fig 3)
- PRAMEL7 recruits CUL2 to chromatin, where CUL2 interacts with a number of chromatin factors, including subunits of NuRD (Fig 4).
- NuRD represses some ground-state genes that are induced upon PRAMEL7 expression (Fig 5). ChIP-seq shows that CHD4 occupancy is decreased upon PRAMEL7 expression (Fig 5).

I feel that the experiments and the logic are rigorous, and they convincingly support the authors' conclusion. The results are novel and interesting. I think this paper would be a good fit for EMBO Reports, after the points below have been taken into consideration.

Minor:

-not immediately clear to me what H/F means in Figure 1. I had to dig in the legend to understand the figure. Also not immediately clear from the figure, the legend, or the results section if the protein is transiently or stably expressed in the ES lines. As it is a critical piece of information, it is not ideal for the reviewer nor for the future reader.

-not clear to me why the authors decided to go with the DeltaN mutant, as opposed to the BC/C2Mut, which seems to also disrupt interaction with CUL2, but is a cleaner mutant as compared to a deletion.

-I am not a fan of the nomenclature used in Fig. 2, with some genes labeled "PRAMEL7 CUL2" and some "PRAMEL7DeltaCUL2". This is confusing and improper on several levels, one of them being that there is no CUL2 deletion. An easier and more logical terminology would be to say that they are CUL2-interaction dependent and CUL2-interaction independent.

-I think Fig 2E is highly interesting. It shows clearly that PRAMEL7 reprograms ES cells. It also shows that PRAMEL7DeltaN has a different effect. Nevertheless, the PCA plot shows clearly that the DeltaN "pushes" the cells to the left in PC1, and also up in PC2. It looks almost as if it triggers the 2i-like transcriptome, but on top of it it also induces/represses other genes. I think it's important to rule out this possibility. It can be done easily, with Venn diagrams, showing how much (or how little) the genes shared by WT and DeltaN overlap with the 2i-specific genes.

-Fig 3C is so small it is unreadable. The authors might as well do away with it.

-Fig. 3F is hard to read, with some of the black or red characters ending on the blue rectangles.

-I do not understand the title on line 274: "PRAMEL7-CUL2 axis contrasts the repression of genes associated with NuRD complex". I guess by "contrasting" they mean "opposing" or "antagonizing"?

Referee #2:

The authors elegantly showed that PRAMEL7 recruits CUL2 to chromatin and targets regulators of repressive chromatin, including NuRD complex for proteasomal degradation, thereby antagonizing NuRD-mediated repression of pluripotency-related genes. These results explain at least one mechanism of how Induced PRAMEL7 expression reprograms developmentally advanced ESC+serum into ground-state pluripotency by inducing a gene expression signature close to developmental ground-state.

However, the following experimental details need to be addressed.

1. Figure 1C, HA immunoprecipitation (IP) in HEK293T cells transfected with HA/FLAG-Pramel7 (H/F-Pramel7) Δ N, why was the full-length PRAMEL7 protein pulled down? In addition, these WB results, especially in the first line, should be shown as results with less exposure.
2. Figure 3C requires enlarged protein names. There is no way to see clearly in the current version.
3. Figure 5E CHD4 ChIP-seq peaks are not convincing. The author needs to show the ChIP-seq input. Had the current "peaks" been compared to the input already?

Referee #3:

Rupasinghe, Bersaglieri and colleagues present results following a previously published paper showing that PRAMEL7 overexpression in mouse embryonic stem cells leads to a change in transcriptome making them closer to the ground-state of pluripotency. Here the authors present a new work showing that PRAMEL7 interacts with CUL2. Combining RNA-sequencing, SILAC-Mass Spectrometry and chromatin immunoprecipitation, they nicely show that the N-terminal domain of PRAMEL7 is involved in CUL2 interaction. These different experiments allow them to conclude that most of the transcriptional changes upon PRAMEL7 overexpression comes from its CUL2-interaction. Further, they propose that PRAMEL7 can bind to chromatin where it recruits CUL2 for subsequent proteasomal degradation of target proteins. They finally suggest that his PRAMEL7-CUL2 interaction regulates the NuRD complex (mainly CHD4) stability and binding to chromatin.

Overall, the manuscript is well-written, and the authors present a compelling story that will be of interest for the field and EMBO Reports readers. I thank the authors for an interesting reading. I therefore recommend this manuscript for publication in EMBO Reports journal following some minor revision. I have one major comment and some minor points listed below.

Major point:

Most of the authors' conclusions on PRAMEL7-CUL2 effect comes from an N-terminal deletion of 40 amino acids. While the results presented are clear, it remains the possibility that the lack of the N-terminal domain itself could partially explain of some of these results. In figures 1B and 1C, the authors show point-mutations of PRAMEL7 that affect interaction with CUL2. Could the authors validate some of their results with one of these constructs (maybe the C2mut)? For example, the Figure 4B with association of PRAMEL7 with chromatin could be done for PRAMEL7 C2mut, but also the change in transcription or protein levels of some of the identified target from Figures 2C and 3E could be confirmed with RT-qPCR and Western blots.

Minor points:

Figure 1C and 1E: the authors present Western blots of PRAMEL7 protein overexpression. A lower band is detectable in the PRAMEL7 WT overexpression below the main signal. Is it some degradation product of PRAMEL7? Could the authors comment on that point? Further on the figure 1E, did the authors check the profiles of PRAMEL7 protein in serum ESC vs 2i-cultured ESC?

Figure 3A: since the level of CHD4 protein is not massively affected (-0.5 log₂ fold-change in Figure 3A), could the author perform an immunofluorescence experiment of CHD4 with or without PRAMEL7 expression to see whether localization pattern is altered? Further on that point, the authors observe a strong decrease of chromatin-bound CHD4 at CHD4-promoters in serum ESC. Could the author identify new CHD4 peaks appearing through the genome? Is it a relocalization of CHD4 to other genomic loci that are now becoming repressed or a release from chromatin as written on line 318? Quantification of CHD4 presence in soluble or chromatin-bound fraction as in Figure 4A could also give an answer to that point.

Figure 3C: I think the authors could split these two diagrams into two separate panels (one for up-regulated and one for down-regulated) to increase visibility.

Figure 5A: the authors show a Western blot of CHD4 protein. In the input line after overexpression of PRAMEL7, a lower band of CHD4 appears that is barely visible without PRAMEL7 expression. Could it be some degradation product? Could the authors check for ubiquitylation or any degradation signal on CHD4 protein or any member of the NuRD complex?

Figure 5B: To my understanding, the authors used an already published dataset (Kloet et al., 2018) for their comparative analysis. However, data presented in Figure 5E-H comes from their own ChIP-seq experiment? Could the authors make this information appear more clearly either on the figure or in the legend?

Line 313: The authors indicate that "32% of genes upregulated in ESC+PRAMEL7 were also significantly regulated upon Mbd3-KD". Could the authors indicate if these 290 genes identified overlap not only with upregulated genes upon PRAMEL7 expression, but also with the 229 genes identified in figure 5B?

Appendix Figure S1C is cited after Appendix Figure S1D in the text. Further, the Appendix Figure S1C is not easy to read. The

histogram on the right would benefit from a legend indicating what blue and red represent. From the text it looks like blue means upregulated genes from PRAMEL7 dataset whereas red represent downregulated? In this case, this is also the opposite from the colors used earlier in the figures (Figures 2A, 2B, 3A, 3B). Further, indicating either in the figure or in the legend from which reference this dataset comes from would be clearer.

Line296: "Appendix Fig. 2A" whereas it should be "1A".

Line303: "Fig. E,F" I believe the "5" is missing.

PRAMEL7/CUL2 axis regulates NuRD stability to establish ground-state pluripotency in embryonic stem cells

Rupasinghe, Bersaglieri *et al.*,

Response to the reviewers

We thank all the Reviewer's for the positive and constructive comments. In the revised manuscript, we have modified the text to clarify points that were not sufficiently clear and included additional data based on the Reviewer's suggestions. The additional data strengthen the conclusions of our work.

List of changes

Modifications in the text were highlighted in red.
The new data are listed here below.

New figures

Revised manuscript	Previous version	Description	Reviewer
Fig. 1C	Fig. 1C	Replacement of better-quality PRAMEL7-CHD4 co-IP	Reviewer 2,3
Fig. 1D	--	WB showing UHRF1 levels upon expression of PRAMEL7 _{WT} and mutants	Reviewer 1,3
Fig. 3C	Fig. 3C	Editing of STRING analysis	Reviewer 1,2
Fig. 3F	Fig. 3F	Style editing of KEGG pathway analysis	Reviewer 1
Fig. 6H	Fig. 5F	Heat map of CHD4-peaks at promoter regions including gained CHD4-peaks in ESC+PRAMEL7	Reviewer 3
EV1A	--	IGV CHD4-ChIPseq with input	Reviewer 2
EV1B	--	Heat map of all CHD4-peaks including gained CHD4-peaks in ESC+PRAMEL7	Reviewer 3
EV1C	--	Average density plots of ChIPseq read counts of CHD4 peaks	Reviewer 3

Additional data not included in the manuscript

Description	Reviewer
Gene expression analysis of "2i-genes"	Reviewer 1
CHD4-immunofluorescence in ESC+PRAMEL7	Reviewer 3

Referee #1:

This paper by Meneka Rupasinghe and co-authors examines mechanisms of pluripotency in mouse ES cells. In an excellent previous paper (Graf NCB 2017), the Santoro lab showed that, in the ICM and in 2i conditions, the protein PRAMEL7 is expressed, causes UHRF1 degradation and hypomethylation, and installs an epiblast-like/ground-state transcription program. This paper also established that PRAMEL7 interacts with CUL2, a RING-type E3 ubiquitin ligase. The current paper builds upon these findings and shows that:

-PRAMEL7 interacts with CUL2 via two consensus boxes in the N terminus (Figure 1). Deleting these boxes renders PRAMEL7 unable to target UHRF1 for degradation.

-As previously described, expression of PRAMEL7 directs ES cells to have a 2i-like transcriptome, whereas the truncated PRAMEL7 does not have the same effect (Figure 2).

-PRAMEL7 controls the protein abundance of many proteins, including a number of chromatin factors such as DNMT1 and NURD. It interacts directly with some of these complexes, as judged by IP/MS using the DeltaN form (Fig 3)

-PRAMEL7 recruits CUL2 to chromatin, where CUL2 interacts with a number of chromatin factors, including subunits of NuRD (Fig 4).

-NuRD represses some ground-state genes that are induced upon PRAMEL7 expression (Fig 5). ChIP-seq shows that CHD4 occupancy is decreased upon PRAMEL7 expression (Fig 5).

I feel that the experiments and the logic are rigorous, and they convincingly support the authors' conclusion. The results are novel and interesting. I think this paper would be a good fit for EMBO Reports, after the points below have been taken into consideration.

Author: We thank the Reviewer for the positive comments.

Minor:

-not immediately clear to me what H/F means in Figure 1. I had to dig in the legend to understand the figure. Also not immediately clear from the figure, the legend, or the results section if the protein is transiently or stably expressed in the ES lines. As it is a critical piece of information, it is not ideal for the reviewer nor for the future reader.

Author: We are sorry for the confusion. At page 5, line 103, as well in the corresponding legend, we defined H/F as the HA/FLAG tag. This abbreviation was because of space restriction in the corresponding figures. We have better clarified in the text that the ESC line stably expressed PRAMEL7_{WT} and PRAMEL7_{ΔN} mutant.

-not clear to me why the authors decided to go with the DeltaN mutant, as opposed to the BC/C2Mut, which seems to also disrupt interaction with CUL2, but is a cleaner mutant as compared to a deletion.

Author: We would like to clarify that PRAMEL7_{ΔN} mutant contains a relatively small deletion (40 aa.), which only comprises the amino acids within the BC and CUL2 boxes domains, leaving intact 83% of PRAMEL7 peptide. We clarified this point in the revised manuscript. The BC/CUL2 point- and deletion-mutants were used to prove that PRAMEL7 contains a functional CUL2 and BC box domains, which are known to mediate the interaction with CUL2, and to assess PRAMEL7 as a CUL2-interacting protein. The data in Figure 1 showed that both BC/CUL2 point- and deletion-mutants have similar phenotype since they cannot interact with CUL2. To better clarify this point, we included additional data (**new Fig. 1D**) further showing that PRAMEL7_{ΔN} and PRAMEL7-point mutants are functionally similar since they are both unable to downregulate the known PRAMEL7-target UHRF1, indicating that both mutants impair PRAMEL7-CUL2 downstream processes.

The reason to perform all the analyses with PRAMEL7_{ΔN} is because we soon obtained an ESC line that had PRAMEL7_{ΔN} expression levels similar to PRAMEL7 in the control ESC+PRAMEL7_{WT}. In contrast, all the generated ESC+PRAMEL7_{BC/C2Mut} cell lines showed very little expression and consequentially could not be used. We do not think there is biology behind, only bad luck. At that time, we decided to initiate the most exciting part of the project (i.e., the functional analyses) with PRAMEL7_{ΔN} cells instead to wait additional time for the ESC+PRAMEL7_{BC/C2Mut} cell lines. Since all the initial experiments indicated

that PRAMEL7 Δ N behave as a CUL2-deficient binding mutant, we continued to use this mutant for all the downstream analyses.

-I am not a fan of the nomenclature used in Fig. 2, with some genes labeled "PRAMEL7 CUL2" and some "PRAMEL7DeltaCUL2". This is confusing and improper on several levels, one of them being that there is no CUL2 deletion. An easier and more logical terminology would be to say that they are CUL2-interaction dependent and CUL2-interaction independent.

Author: We agree with the Reviewer. We modified the nomenclature accordingly (i.e., CUL2-interaction dependent genes \rightarrow short form "PRAMEL7_{CUL2}" and CUL2-interaction independent genes \rightarrow short form "PRAMEL7_{CUL2-ind}").

-I think Fig 2E is highly interesting. It shows clearly that PRAMEL7 reprograms ES cells. It also shows that PRAMEL7DeltaN has a different effect. Nevertheless, the PCA plot shows clearly that the DeltaN "pushes" the cells to the left in PC1, and also up in PC2. It looks almost as if it triggers the 2i-like transcriptome, but on top of it it also induces/represses other genes. I think it's important to rule out this possibility. It can be done easily, with Venn diagrams, showing how much (or how little) the genes shared by WT and DeltaN overlap with the 2i-specific genes.

Author: As described in our previous work, ESC+PRAMEL7 and ESC+2i act on different pathways and, although both similar to ground-state pluripotency, they do not share very similar transcription profiles (Graf et al., 2017). We followed the suggestion of the Reviewer and analyzed the expression of genes common to PRAMEL7_{WT} and PRAMEL7 Δ N relative to 2i- and serum-specific genes reported by Marks et al. 2012 (PMID: 22541430, see data below). We have also analyzed our own RNAseq in ESC+2i and ESC+serum (Dalcher et al., PMID: 33433018). However, whatever dataset we used, we found very little overlap, indicating once more that ESC+PRAMEL7 are similar to early embryo pluripotency but not to ESC+2i. We decided to not include these data in the revised manuscript since they were not informative or novel.

-Fig 3C is so small it is unreadable. The authors might as well do away with it.

Author: We increased the size of the protein names.

-Fig. 3F is hard to read, with some of the black or red characters ending on the blue rectangles.

Author: We modified the style of Figure 3F.

-I do not understand the title on line 274: "PRAMEL7-CUL2 axis contrasts the repression of genes associated with NuRD complex". I guess by "contrasting" they mean "opposing" or "antagonizing"?

Author: We thank the reviewer for this comment. We modified the title accordingly.

Referee #2:

The authors elegantly showed that PRAMEL7 recruits CUL2 to chromatin and targets regulators of repressive chromatin, including NuRD complex for proteasomal degradation, thereby antagonizing NuRD-mediated repression of pluripotency-related genes. These results explain at least one mechanism of how Induced PRAMEL7 expression reprograms developmentally advanced ESC+serum into ground-state pluripotency by inducing a gene expression signature close to developmental ground-state.

Author: We thank the Reviewer for the positive comments.

However, the following experimental details need to be addressed.

1. Figure 1C, HA immunoprecipitation (IP) in HEK293T cells transfected with HA/FLAG-Pramel7 (H/F-Pramel7) Δ N, why was the full-length PRAMEL7 protein pulled down? In addition, these WB results, especially in the first line, should be shown as results with less exposure.

Author: We apologize for the quality of this IP. We replaced this experiment with a new IP which confirms the previous results showing that all PRAMEL7 BC/CUL2 box mutants do not associate with CUL2.

2. Figure 3C requires enlarged protein names. There is no way to see clearly in the current version.

Author: We increased the size of the protein names.

3. Figure 5E CHD4 ChIP-seq peaks are not convincing. The author needs to show the ChIP-seq input. Had the current "peaks" been compared to the input already?

Author: We are sorry for the confusion. As indicated in the M&M section, CHD4 bound regions were defined using SICER by comparing the CHD4 ChIPs of the three cell lines and the respective inputs. We did not include the input signal in the IGV image of Figure 5E (**now Fig. 6G**) to simplify the visualization of the ChIPseq and RNAseq data. In this revised version, we included IGV images of the ChIPseq containing the corresponding inputs (**Appendix Fig. 1A**).

Referee #3:

Rupasinghe, Bersaglieri and colleagues present results following a previously published paper showing that PRAMEL7 overexpression in mouse embryonic stem cells leads to a change in transcriptome making them closer to the ground-state of pluripotency. Here the authors present a new work showing that PRAMEL7 interacts with CUL2. Combining RNA-sequencing, SILAC-Mass Spectrometry and chromatin immunoprecipitation, they nicely show that the N-terminal domain of PRAMEL7 is involved in CUL2 interaction. These different experiments allow them to conclude that most of the transcriptional changes upon PRAMEL7 overexpression comes from its CUL2-interaction. Further, they propose that PRAMEL7 can bind to chromatin where it recruits CUL2 for subsequent proteasomal degradation of target proteins. They finally suggest that his PRAMEL7-CUL2 interaction regulates the NuRD complex (mainly CHD4) stability and binding to chromatin.

Overall, the manuscript is well-written, and the authors present a compelling story that will be of interest for the field and EMBO Reports readers. I thank the authors for an interesting reading. I therefore recommend this manuscript for publication in EMBO Reports journal following some minor revision.

Author: We thank the Reviewer for the positive comments.

I have one major comment and some minor points listed below.

Major point:

Most of the authors' conclusions on PRAMEL7-CUL2 effect comes from an N-terminal deletion of 40 amino acids. While the results presented are clear, it remains the possibility that the lack of the N-terminal domain itself could partially explain of some of these results. In figures 1B and 1C, the authors show point-mutations of PRAMEL7 that affect interaction with CUL2. Could the authors validate some of their results with one of these constructs (maybe the C2mut)? For example, the Figure 4B with association of PRAMEL7 with chromatin could be done for PRAMEL7 C2mut, but also the change in transcription or protein levels of some of the identified target from Figures 2C and 3E could be confirmed with RT-qPCR and Western blots.

Author: We are sorry for the confusion. We would like to clarify that PRAMEL7 Δ N mutant contains a relatively small deletion (40 aa.), which only comprises the amino acids within the BC/CUL2 domain, leaving intact 83% of PRAMEL7 peptide. We included this information in the results section of this revised work. The PRAMEL7 deletion- and point-mutants were analyzed to demonstrate that they do not interact with CUL2 and to assess that PRAMEL7 contains a functional BC/CUL2 domain, which is known to mediate the interaction with CUL2, making PRAMEL7 a CUL2-interacting protein. Since BC/CUL2 domain is specifically recognized by CUL2, we think that it is quite unlikely that the small and precise deletion of the BC/CUL2 domain would affect the association of PRAMEL7 with factors other than the CUL2 complex in particularly since all the data showed a role of CUL2 in PRAMEL7-mediated regulation, including the interaction of CUL2 with CHD4 on chromatin which is PRAMEL7-dependent. To further demonstrate that PRAMEL7 Δ N and ESC+PRAMEL7_{BC/C2Mut} have similar function we included new data showing that both mutants cannot downregulate UHRF1, a known PRAMEL7 target (**new Fig. 1D**). We could not perform additional comparative experiments since we fail to establish ESC lines expressing ESC+PRAMEL7_{BC/C2Mut}.

Minor points:

Figure 1C and 1E: the authors present Western blots of PRAMEL7 protein overexpression. A lower band is detectable in the PRAMEL7 WT overexpression below the main signal. Is it some degradation product of PRAMEL7? Could the authors comment on that point?

Author: We replaced Figure 1C with a co-IP of better quality which does not show any band below PRAMEL7 signal. We did not observe this band in any WBs.

Further on the figure 1E, did the authors check the profiles of PRAMEL7 protein in serum ESC vs 2i-cultured ESC?

Author: We performed this analysis in our previous work (Graf et al., 2017). PRAMEL7 protein levels in ESC+2i and ESC+serum are undetectable by WB. At mRNA level, however, PRAMEL7 levels are higher in ESC+2i than in ESC+serum.

Figure 3A: since the level of CHD4 protein is not massively affected (-0.5 log₂ fold-change in Figure 3A), could the author perform an immunofluorescence experiment of CHD4 with or without PRAMEL7 expression to see whether localization pattern is altered?

Author: We performed CHD4-IFs in ESCs and ESC+PRAMEL7. However, we did not observe any evident changes in signals. We decided to not include this data in the manuscript since they were not informative.

Further on that point, the authors observe a strong decrease of chromatin-bound CHD4 at CHD4-promoters in serum ESC. Could the author identify new CHD4 peaks appearing through the genome? Is it a relocalization of CHD4 to other genomic loci that are now becoming repressed or a release from chromatin as written on line 318? Quantification of CHD4 presence in soluble or chromatin-bound fraction as in Figure 4A could also give an answer to that point.

Author: We reanalysed the CHD4-ChIPseq for all peaks and peaks within promoter (**new Figs. 6H and Appendix Fig. 1B**). We found that only few sites gained CHD4 signal in ESC+H/F-PRAMEL7_{WT}, indicating that the expression of PRAMEL7 did not cause a relocalization of CHD4 but instead promoted a loss of CHD4 association with chromatin. Accordingly, the few promoters (100), which become bound by CHD4 in ESC+PRAMEL7, showed a weak CHD4 signal compared to average CHD4 signal in control cells (**new Appendix Fig. 1C**) and in general they were not significantly downregulated (only two genes were downregulated).

Figure 3C: I think the authors could split these two diagrams into two separate panels (one for up-regulated and one for down-regulated) to increase visibility.

Author: We divided Figure 3C in two parts (now. Fig. 3C and D).

Figure 5A: the authors show a Western blot of CHD4 protein. In the input line after overexpression of PRAMEL7, a lower band of CHD4 appears that is barely visible without PRAMEL7 expression. Could it be some degradation product? Could the authors check for ubiquitinylation or any degradation signal on CHD4 protein or any member of the NuRD complex?

Author: We increased the exposure of the indicated WB (see image below), however, we did not find any evident difference in CHD4 signals between parental ESCs and ESC+PRAMEL7. We followed the suggestion of the Reviewer and performed western blot for ubiquitin on CHD4-IPs from parental ESCs and ESC+PRAMEL7. However, we did not observe any difference. These results are in agreement with our data showing that PRAMEL7-CUL2-mediated downregulation of CHD4 can be detected only with quantitative proteomic analyses but not with western blot.

Figure 5B: To my understanding, the authors used an already published dataset (Kloet et al., 2018) for their comparative analysis. However, data presented in Figure 5E-H comes from their own ChIP-seq experiment? Could the authors make this information appear more clearly either on the figure or in the legend?

Author: We included this information in the corresponding Figure legend.

Line 313: The authors indicate that "32% of genes upregulated in ESC+PRAMEL7 were also significantly regulated upon Mbd3-KD". Could the authors indicate if these 290 genes identified overlap not only with upregulated genes upon PRAMEL7 expression, but also with the 229 genes identified in figure 5B?

Author: We did this analysis and found that 21% of genes that are upregulated in ESC+PRAMEL7 and in *Mbd3*-KD cells contain CHD4 bound to their promoters and are significantly linked to cell signalling pathways ($P 10^{-6}$). Although this is a nice correlation, we would like to not include this information in the revised manuscript. Indeed, the analysis identifying PRAMEL7-regulated genes with CHD4-bound promoters (**Fig. 5B**) was very stringent since it excluded genes that could be regulated by CHD4 bound at probable enhancers and coding regions.

Appendix Figure S1C is cited after Appendix Figure S1D in the text.

Author: We have reorganized Appendix Figure S1 and the corresponding citations in the text.

Further, the Appendix Figure S1C is not easy to read. The histogram on the right would benefit from a legend indicating what blue and red represent. From the text it looks like blue means upregulated genes from PRAMEL7 dataset whereas red represent downregulated? In this case, this is also the opposite from the colors used earlier in the figures (Figures 2A, 2B, 3A, 3B). Further, indicating either in the figure or in the legend from which reference this dataset comes from would be clearer.

Author: We included the missing information in the Figure and the corresponding legend.

Line296: "Appendix Fig. 2A" whereas it should be "1A".

Line303: "Fig. E,F" I believe the "5" is missing.

Author: Thank you for having spotting these mistakes. We corrected these typos, accordingly.

Dear Raffaella, and happy new year !

Thank you for the submission of your revised manuscript. We have now received the enclosed reports from the referees that were asked to assess it and I am happy to say that both support the publication of your work now. Only a few more minor editorial requests will need to be addressed before we can proceed with the official acceptance of your manuscript.

- Please rename the conflict of interest subheading to "Disclosure and Competing Interest Statement".
- Please remove the author credits from the ms file. All credits need to be entered online during ms submission.
- Please send us a completed author checklist that can be found here: <https://www.embopress.org/page/journal/14693178/authorguide>. The completed list will also be part of our transparent peer-review process file.
- You uploaded 9 Datasets; their legends need to be removed from the ms and placed in each Excel file, as a separate sheet/tab; while the callouts in the ms are OK, the files themselves need to be renamed: e.g. it should be "Dataset EV1" instead of "Table S1".
- The Appendix file needs a title page that has a table of content with page numbers; the figure needs "S" in the name, it should be Appendix Figure S1; the callout in the ms needs to be updated too; its legend needs to be removed from the ms file and placed in the Appendix file, right after the figure.
- The source data (SD) need to be uploaded as one folder per one figure, the folder for Figure 6 also contains SD for figure 5, these need to be split.
- Summary should be renamed to Abstract.
- Our routine image analysis flagged a possible re-use of the heatmap image between Figure 6h and Appendix figure 1B, and this is not listed in figure legend. Can you please check and clarify? Thank you.
- Please add direct links in the DAS for the GSE215339 and PXD032184 datasets.
- Please specify the statistical tests used for data analysis in the legends of figures 2a-b, d; 3a-b; 6e.

I would like to suggest some minor changes to the title and abstract. Please let me know whether you agree with the following:

PRAMEL7 and CUL2 decrease NuRD stability to establish ground-state pluripotency

Pluripotency is established in the E4.5 preimplantation epiblast. Embryonic stem cells (ESCs) represent both immortalization and pluripotency, however, their gene expression signature only partially resembles that of developmental ground-state. Induced PRAMEL7 expression, a protein highly expressed in the ICM but lowly expressed in ESCs, reprograms developmentally advanced ESC+serum into ground-state pluripotency by inducing a gene expression signature close to developmental ground-state. However, how PRAMEL7 reprograms gene expression remains elusive. Here we show that PRAMEL7 associates with Cullin2 (CUL2) and this interaction is required to establish ground-state gene expression. PRAMEL7 recruits CUL2 to chromatin and targets regulators of repressive chromatin, including the NuRD complex, for proteasomal degradation. PRAMEL7 antagonizes NuRD-mediated repression of genes implicated in pluripotency by decreasing NuRD stability and promoter association in a CUL2-dependent manner. Our data link proteasome degradation pathways to ground-state gene expression, offering insights to generate in vitro models to reproduce in vivo ground-state pluripotency.

EMBO press papers are accompanied online by A) a short (1-2 sentences) summary of the findings and their significance, B) 2-3 bullet points highlighting key results and C) a synopsis image that is exactly 550 pixels wide and 200-600 pixels high (the height is variable). You can either show a model or key data in the synopsis image. Please note that text needs to be readable at the final size. Please send us this information along with the final manuscript.

Best wishes,
Esther

Esther Schnapp, PhD
Senior Editor

EMBO reports

Referee #1:

I thank the authors for their modifications, which fully answer my questions.

Referee #3:

The authors have addressed my comments adequately and I therefore recommend this manuscript for publication in EMBO Reports.

All editorial and formatting issues were resolved by the authors.

Prof. Raffaella Santoro
University of Zurich
Molecular Mechanisms of Disease
Winterthurerstrasse 190
Zurich 8057
Switzerland

Dear Raffaella,

I am very pleased to accept your manuscript for publication in the next available issue of EMBO reports. Thank you for your contribution to our journal.

Best wishes,
Esther
